# DiffPO: A causal diffusion model for learning distributions of potential outcomes

**Yuchen Ma, Valentyn Melnychuk, Jonas Schweisthal & Stefan Feuerriegel**
LMU Munich
Munich Center for Machine Learning
`yuchen.ma@lmu.de`

## Abstract

Predicting potential outcomes of interventions from observational data is crucial for decision-making in medicine, but the task is challenging due to the fundamental problem of causal inference. Existing methods are largely limited to point estimates of potential outcomes with no uncertain quantification; thus, the full information about the distributions of potential outcomes is typically ignored. In this paper, we propose a novel causal diffusion model called DiffPO, which is carefully designed for reliable inferences in medicine by learning the *distribution* of potential outcomes. In our DiffPO, we leverage a tailored conditional denoising diffusion model to learn complex distributions, where we address the selection bias through a novel orthogonal diffusion loss. Another strength of our DiffPO method is that it is highly flexible (e.g., it can also be used to estimate different causal quantities such as CATE). Across a wide range of experiments, we show that our method achieves state-of-the-art performance.

## 1 Introduction

Predicting potential outcomes (POs) for patients from observational data is crucial for decision-making in medicine [16]. For example, in cancer care, one is interested in individualized predictions of survival under different treatment plans, which can then help medical practitioners choose a treatment plan that promises the largest chance of survival [48].

Predicting POs of interventions from observational data is challenging due to the *fundamental problem of causal inference*, i.e., the fact that one cannot obverse all POs for each individual [27]. Further, in observational data, treatments are generally not assigned completely randomly, and, as a result, the distributions of covariates in the treatment and control groups differ [54]. If this covariate shift is not taken into account, the distributions of POs given the covariates would be learned sub-optimally [13, 11].

In the causal inference literature, many existing methods are aimed at conditional average treatment effect (CATE) estimation where POs are used as auxiliary quantities (e.g.,[1, 2, 3, 6, 10, 12, 13, 25, 28, 30, 31, 32, 37, 45, 64, 66]). In principle, some CATE methods *could* be used for predicting POs but these are not tailored for POs and thus tend to *underperform* at predicting POs. The underlying reason is that the estimator with the best performance in estimating CATE might not do best at predicting POs in finite samples [11, 13]. Additionally, many state-of-the-art CATE estimators leverage an inductive bias [13, 31], i.e., they assume that POs should share similar structures. As a result, they assume that the CATE follows a much simpler function than the POs, so that it is easier to learn the CATE than each PO separately. Moreover, most of these methods only focus on point estimates instead of learning the distributions of POs. The latter is more difficult but very crucial for reliable decision-making in medical applications.

38th Conference on Neural Information Processing Systems (NeurIPS 2024).

Table 1: Overview of key methods for CATE estimation (and predicting POs). *UQ* refers to whether the methods allow for uncertainty quantification of POs. *Bias addressing* refers to whether the methods address the selection bias. *Orthogonal* refers to whether it is robustness (i.e., Neyman-orthogonality wrt. nuisance functions). *POs are the target* refers to whether the methods are originally designed for the task of predicting POs.

| | Estimator type | UQ | Bias-addressing | Orthogonal | POs are the target | Key limitations |
|---|---|---|---|---|---|---|
| S-learner [37] | One-step (plug-in), model-agnostic | ✗ | ✗ | ✗ | ✓ | Selection bias; point estimator |
| T-learner [37] | One-step (plug-in), model-agnostic | ✗ | ✗ | ✗ | ✓ | Selection bias; point estimator |
| DR-learner[31, 39] | Two-step, model-agnostic | ✗ | ✓ | ✓ | ✗ | CATE inductive bias; point estimator |
| RA-learner[12] | Two-step, model-agnostic | ✗ | ✓ | ✗ | ✗ | CATE inductive bias; point estimator |
| TARNet [54] | One-step (plug-in), model-specific | ✗ | ✗ | ✗ | ✓ | Selection bias; point estimator |
| CFR [30] | One-step (plug-in), model-specific | ✗ | ✓ | ✗ | ✓ | Point estimator |
| GANITE [64] | Two-step, model-specific | (✗) | ✗ | ✗ | (✗) | Selection bias; unstable training |
| TEDVAE [66] | One-step (plug-in), model-specific | ✗ | ✗ | ✗ | ✗ | Selection bias; non-identifiability |
| **DiffPO** (ours) | One-step (plug-in), model-specific | ✓ | ✓ | ✓ | ✓ | Sampling time longer |

(✗) in the column *UQ* refers to that methods are not originally designed for uncertainty quantification of POs, but can be adapted for this purpose.

In this paper, we propose a causal diffusion model called DiffPO for predicting POs. Specifically, we leverage a tailored conditional diffusion model to learn complex distributions of POs. We further propose a novel orthogonal diffusion loss to adjust for the covariate shift problem in finite samples. Our orthogonal diffusion loss ensures Neyman-orthogonality, which offers favorable theoretical properties. In particular, it makes our method more robust to misspecification. Further, our DiffPO is carefully designed for *reliable* inferences in medicine and thus allows for uncertainty quantification We thereby follow recent calls in medicine to move beyond point estimates and offer distributional knowledge [23, 35], which can inform how likely a certain outcome is and what a probable range of POs is.

Our method is highly flexible. It can learn distributions of POs for uncertainty quantification and give point estimates of POs. It can also handle different causal quantities, such as the CATE. This is unlike many state-of-the-art methods from CATE estimation which are limited to point estimates [12, 13, 31, 32, 37, 45]. Hence, for different settings, custom methods have to be designed, while our causal diffusion model offers a flexible and reliable approach.

Overall, our **main contributions** are the following: (1) We propose a novel causal diffusion model for learning the distributions of POs, which can give both point estimation but also allows for uncertainty quantification. (2) We propose a novel orthogonal diffusion loss that ensures Neyman-orthogonality. (3) We conduct a wide range of experiments in different settings to demonstrate the flexibility and effectiveness of our DiffPO.[1]

## 2 Related Work

Our work aims to learn the distributions of POs for patients. The task is related to CATE estimation in that the PO framework conceptualizes the CATE as the expected difference between the POs of a patient with and without treatment. Hence, we include also methods for CATE estimation in our literature review below. While these could, in principle, be used for predicting POs, *we emphasize that CATE estimation and predicting POs will have a different finite-sample performance.*

### 2.1 Predicting POs and CATE estimation

There have been many works designed for CATE estimation (e.g.,[1, 2, 3, 6, 10, 12, 13, 25, 28, 30, 31, 32, 37, 45, 64, 66]). Existing CATE methods fall into two categories: model-agnostic and model-specific estimators. We only give a concise overview in this section (see Table 1). A detailed literature review is in Appendix A.3.

**Model-agnostic estimators.** Model-agnostic learning strategies are also known as meta-learners (e.g.,[12, 13, 31, 32, 37, 45]). These can be split into (a) one-step (plug-in) learners that output regression functions from the observational data and then compute the CATE as the difference between the POs and (b) two-step learners that first estimate nuisance functions to build a pseudo-outcome and then obtain CATE directly by regressing the input covariates on the pseudo-outcomes in the second step. (Note that *pseudo-outcomes are not potential outcomes*).

---

[1] Code is available at `https://github.com/yccm/DiffPO`.

The problem with one-step plug-in learners (e.g., S-learner [37], T-learner [37]) is that they often do not address selection bias. The problem with two-step learners (e.g., DR-learner [31, 39], RA-learner [12]) is that these commonly leverage an inductive bias specific to CATE. Thus, two-step learners often have benefits for CATE estimation, but they make restrictive assumptions that hurt the finite-sample performance in predicting POs. Hence, in finite samples, such an inductive bias in CATE estimation can even lead to bias in PO prediction.

**Model-specific estimators.** Many model-specific estimators provide instantiations the general model-agnostic methods: Here, standard machine learning models are adapted to CATE estimation / POs prediction (e.g.,[10, 30, 54, 61, 64, 66]). Recently, neural networks have been used for model-specific learners [30, 54, 64, 66]. Yet, model-specific estimators based on neural networks are designed for CATE estimation but are *not* directly aimed at POs.

**Uncertainty quantification.** Most state-of-the-art estimators fail to offer uncertainty quantification of POs (see Table 1). In particular, these methods typically only offer point estimates, yet *without* distributional information and thus do *not* allow for uncertainty quantification. Yet, distributional information is crucial for reliable decision-making in medicine and presents the focus of our method.

**Implications for benchmarking.** Most of the existing works are designed for CATE estimation, but not for learning the distributions of POs. For benchmarking PO prediction, many methods targeting the CATE *do not include an estimation of POs*, and, thus, they are *not* applicable for our task. We show this in the column *POs are the target* in Table 1. Instead, we can *only* compare with those methods that are applicable to our task: *that is, where the model can directly output POs predictions* [37, 54, 64].

## 2.2 Diffusion models for causal inference

Diffusion models were introduced for learning from complex distribution for a given dataset from which one can then sample high-quality data points (e.g., images) [26, 55, 57]. Diffusion models have achieved state-of-the-art performance, outperforming other generative models on various tasks in the computer vision field (e.g.,[14, 56, 59]). We give a brief technical overview of diffusion models in Sec. 3.

Diffusion models were previously used for different causal inference tasks but in a *different* setting from ours, for example, generating the counterfactual of a given image [36, 51], answering causal queries [8], or causal discovery [52]. We emphasize that the aforementioned tasks are different from ours. For example, the task in [36, 51] is similar to minimizing the changes that one needs to make to an image in order for the classifier to categorize the image into a different class. The task of answering causal queries [8, 33, 53] builds upon structural causal models (SCMs), while we are using the PO framework [50]. The assumptions for fitting SCMs are usually much stronger than the latter. In causal discovery [52], the output is the causal graph, while our output is a causal quantity. Hence, these works were all developed for *different* tasks (and *not* designed for learning the distributions of POs). Because of this, the works are *not* applicable as baselines.

## 3 Preliminaries on diffusion models

Diffusion models [26, 55, 57] are likelihood-based generative models that use (1) forward and (2) reverse Markov processes. The (1) **forward process** 'disturbs' the data distribution $q(x_0)$ into a tractable prior $\mathcal{N}(\mathbf{0}, \mathbf{I})$. For this, it gradually adds noise to an initial sample $x_0 \sim q(x_0)$ across $T$ steps with variance schedules $\{\beta_1, \ldots, \beta_T\}$. The forward process can be written as $q(x_{1:T} \mid x_0) = \prod_{t=1}^{T} q(x_t \mid x_{t-1})$ with Gaussian distribution $q(x_t \mid x_{t-1}) := \mathcal{N}(x_t; \sqrt{1 - \beta_t} x_{t-1}, \beta_t \mathbf{I})$. It admits a closed form of $q(x_t \mid x_0)$ that is also a Gaussian distribution $\mathcal{N}(\sqrt{\bar{\alpha}_t} x_0, (1 - \bar{\alpha}_t) \mathbf{I})$, where $\bar{\alpha}_t = \prod_{i=1}^{t} (1 - \beta_i)$.

The (2) **reverse process** $p(x_{0:T}) = \prod_{t=1}^{T} p(x_{t-1} \mid x_t)$ gradually denoises the latent variable $x_T \sim \mathcal{N}(\mathbf{0}, \mathbf{I})$ and further allows for generating new data samples from $q(x_0)$. The distributions $p(x_{t-1} \mid x_t)$ are usually unknown and approximated by a neural network with parameters $\theta$. Thus, a parameterized Markov chain $\{p_\theta(x_{t-1} \mid x_t)\}_{t=1}^{T}$ is trained in the reverse process. It can be parameterized as $p_\theta(x_{t-1} \mid x_t) := \mathcal{N}(x_{t-1}; \mu_\theta(x_t, t), \Sigma_\theta(x_t, t))$. [26] suggested using diagonal $\Sigma_\theta(x_t, t)$ with a constant $\sigma_t$ and computing $\mu_\theta(x_t, t)$ as a function of $x_t$ and $\epsilon_\theta(x_t, t)$. This yields

$\mu_\theta(x_t, t) = \frac{1}{\sqrt{\alpha_t}} \left( x_t - \frac{\beta_t}{\sqrt{1-\bar{\alpha}_t}} \epsilon_\theta(x_t, t) \right)$, where $\alpha_t := 1 - \beta_t, \bar{\alpha}_t := \prod_{i \leq t} \alpha_i$ and where $\epsilon_\theta(x_t, t)$ predicts a 'ground truth' noise component $\epsilon$ for the noisy data sample $x_t$.

Later, we develop a novel causal diffusion model for predicting POs. We further explain why the standard loss from above fails in our task because it does *not* account for the underlying causal structure. As a remedy, we develop an orthogonal diffusion loss that is tailored to learn complex distributions of POs.

## 4 Problem Formulation

**Setup:** We consider an observational dataset $\mathcal{D}$ with i.i.d. patient data. The dataset consists of: an outcome of interest $Y \in \mathcal{Y} \subseteq \mathbb{R}$, $d_X$-dimensional covariates (also called confounders) $X \in \mathcal{X} \subseteq \mathbb{R}^{d_X}$, and a treatment $A \in \{0, 1\}$. For example, in critical care, the patient covariates $X$ are different risk factors (e.g., age, gender, prior diseases), the treatment is whether a ventilator is applied, and the outcome is the patient survival. For notation, let $\pi(x) = p(A = 1 \mid X = x)$ denote the propensity score, which gives the probability of a patient receiving the treatment. Let $p(y \mid x, a) = p(Y = y \mid X = x, A = a)$ be the probability density function of the conditional distribution $p(Y \mid X, A)$.

**Potential outcomes:** We build upon the standard setting of Neyman-Rubin potential outcomes framework [50]. Hence, let $Y(a)$ denote the *potential outcome* after intervening on the treatment by setting it to $a$. We have two potential outcomes for each individual: $Y(1)$ if treatment is administered (i.e., $A = 1$), and $Y(0)$ if not treated (i.e., $A = 0$). However, due to the fundamental problem of causal inference [27], only one of the POs is observed. Hence, $Y = AY(1) + (1 - A)Y(0)$.

**Identifiability:** To ensure that POs are identifiable, we follow previous literature (e.g.,[12, 31, 64]) and make the following standard assumptions:

**Assumption 1.** *(1) Consistency: If an individual is assigned treatment $a$, we observe the associated potential outcome $Y = Y(a)$. (2) Unconfoundedness: there are no unobserved confounders, so that $Y(0), Y(1) \perp\!\!\!\perp A \mid X$. (3) Overlap: treatment assignment is non-deterministic, i.e., $0 < \pi(x) < 1, \forall x \in \mathcal{X}$ if $p(x) > 0$.*

Under Assumption 1, the distributions of POs can be identified from observational data via $p(Y(a) \mid X) = p(Y \mid X, A = a)$.

**Objective:** Our main interest lies in *predicting POs in medical settings*. Formally, $\mathbb{E}[Y(a) \mid X]$ are the expected POs corresponding to a treatment assignment (intervention) $a$ for an individual with covariates $X$. Predicting POs is crucial for decision support in medicine [16]. For example, in critical care, doctors aim to predict the survival of each patient under different treatments (e.g., mechanical ventilation), which can then guide their decision-making.

However, *decision-making in medicine needs to be reliable*. Because of this, doctors are not only interested in the point estimate but need to *quantify the uncertainty in the POs* [23, 35]. For example, uncertainty quantification is needed in medical practice to decide whether to apply either a treatment that has a small but certain benefit or a treatment with a large benefit but potentially a large variability in the outcomes and thus large uncertainty of whether the treatment is actually beneficial for a specific patient. Hence, we estimate the *distribution* of the POs after assigning treatment $a$ to an individual, i.e., $p(Y(a) \mid X)$. Learning this distribution allows us to sample from it and obtain predictive intervals for uncertainty quantification.

## 5 DiffPO for predicting potential outcomes

**Overview**: In the following, we introduce our DiffPO: a diffusion-based model for predicting POs with 3 key components (Fig. 1): ① a *forward diffusion process*, ② a *reverse diffusion process*, and ③ a novel *orthogonal diffusion loss*. Finally, we introduce our training and sample procedure.

## 5.1 Forward and reverse diffusion process

Our DiffPO builds upon diffusion models [26, 55, 57]. We distinguish the distributions learned in the forward process and the reverse process via $q$ and $p$, respectively, to make the notation straightforward in this section.

① **Forward process:** Given a data point $(y, x, a)$ sampled from the observational data distribution $p(Y, X, A)$, in the forward diffusion process, we gradually add Gaussian noise to the initial $y$ (denoted as $y_0$) across $T$ time steps. This thus produces a sequence of noisy samples $y_1, \ldots, y_T$ in the same sample space as $y_0$. The forward process follows a Markov chain

$$q\left(y_{1:T} \mid y_0\right) := \prod_{t=1}^{T} q\left(y_t \mid y_{t-1}\right), \quad q\left(y_t \mid y_{t-1}\right) := \mathcal{N}\left(y_t; \sqrt{1 - \beta_t} y_{t-1}, \beta_t \mathbf{I}\right), \tag{1}$$

where $\beta_t$ is a variance schedule: $\{\beta_t \in (0, 1)\}_{t=1}^{T}$ is a small positive constant that represents a noise level and, thus, essentially controls the step sizes. By using the reparameterization trick [34], sampling $y_t$ at an arbitrary timestep $t$ has a closed form

$$q\left(y_t \mid y_0\right) = \mathcal{N}\left(y_t; \sqrt{\bar{\alpha}_t} y_0, (1 - \bar{\alpha}_t) \mathbf{I}\right), \tag{2}$$

where $\alpha_t := 1 - \beta_t$ and $\bar{\alpha}_t := \prod_{s=1}^{t} \alpha_s$. Thus, $y_t$ can be expressed as

$$y_t\left(y_0, \epsilon\right) = \sqrt{\bar{\alpha}_t} y_0 + \sqrt{1 - \bar{\alpha}_t} \epsilon, \tag{3}$$

where $\epsilon \in \mathbb{R}^{d_Y} \sim \mathcal{N}(\mathbf{0}, \mathbf{I})$. As a result, the data sample $y_0$ gradually 'looses' its distinguishable features for later steps $t$. Eventually, when $T \to \infty$, $y_T$ is equivalent to an isotropic Gaussian distribution.

② **Reverse process:** We aim to learn the true distributions of POs, which can be identified as the conditional distribution in Sec. 4. Using the notation above, it can be rewritten as $q\left(y_0 \mid x, a\right) = \int q\left(y_{0:T} \mid x, a\right) \mathrm{d}y_{1:T}$. As $q\left(y_{t-1} \mid y_t, x, a\right)$ is intractable, we need to learn a model distribution $p_\theta$ parameterized by $\theta$ to approximate the data distribution.

The reverse process starts at a known prior distribution with density $p\left(y_T\right) = \mathcal{N}\left(y_T; \mathbf{0}, \mathbf{I}\right)$ and then proceeds backward to obtain the data distribution at step $t = 0$. Formally, the reverse diffusion process is also a Markov chain, and the density of its joint distribution $p_\theta\left(y_{0:T} \mid x, a\right)$ can be written as

$$p_\theta\left(y_{0:T} \mid x, a\right) := p\left(y_T\right) \prod_{t=1}^{T} p_\theta\left(y_{t-1} \mid y_t, x, a\right), \quad y_T \sim \mathcal{N}(\mathbf{0}, \mathbf{I}). \tag{4}$$

We employ a conditional diffusion model with the reverse process in Eq. (4). The conditional density $p_\theta\left(y_{t-1} \mid y_t, x, a\right)$ is also Gaussian, and we parameterize it as

$$p_\theta\left(y_{t-1} \mid y_t, x, a\right) := \mathcal{N}\left(y_{t-1}; \mu_\theta\left(y_t, t \mid x, a\right), \sigma_t^2 \mathbf{I}\right). \tag{5}$$

The reverse process gradually denoises $y_T \sim \mathcal{N}(\mathbf{0}, \mathbf{I})$ through the learned Gaussian transitions. Once we have computed the latter, we approximate the original data distribution and can even sample from it.

**Variational inference:** Directly computing and maximizing the likelihood $p_\theta(y_0 \mid x, a)$ is difficult because it involves intractable integration. As a remedy, we frame the learning objective for the above diffusion processes through variational inference. Thus, the parameters $\theta$ can then be optimized by maximizing the evidence lower bound (ELBO) via

$$\log p_\theta\left(y_0 \mid x, a\right) \geq \mathbb{E}_{y_{1:T} \sim q(Y_{1:T} \mid y_0)} \left[\log \frac{p_\theta\left(y_{0:T} \mid x, a\right)}{q\left(y_{1:T} \mid y_0\right)}\right]. \tag{6}$$

The right side of Eq. (6) can be rewritten as (see Appendix B for the derivation)

$$\underbrace{\mathbb{E}_{y_1 \sim q(Y_1 \mid y_0)} \left[\log p_\theta\left(y_0 \mid y_1, x, a\right)\right]}_{\text{reconstruction term}} - \underbrace{D_{\mathrm{KL}}\left(q\left(y_T \mid y_0\right) \| p\left(y_T\right)\right)}_{\text{prior matching term}}$$

$$- \sum_{t=2}^{T} \underbrace{\mathbb{E}_{y_t \sim q(Y_t \mid y_0)} \left[D_{\mathrm{KL}}\left(q\left(y_{t-1} \mid y_t, y_0\right) \| p_\theta\left(y_{t-1} \mid y_t, x, a\right)\right)\right]}_{\text{denoising matching term}}. \tag{7}$$

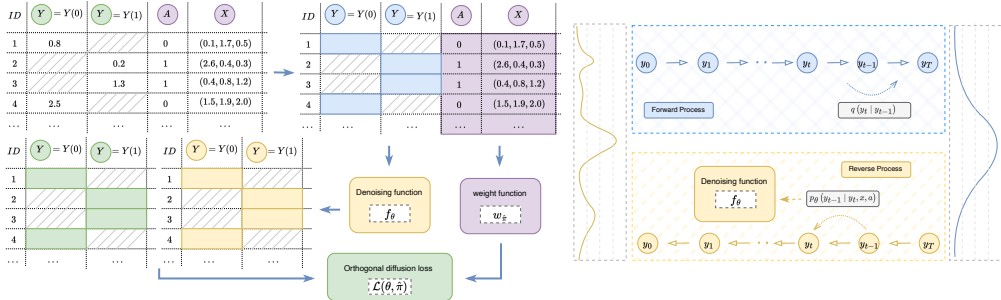

Figure 1: **Overview of our causal diffusion model DiffPO.** Our method involves a forward and reverse diffusion process to learn the distributions of potential outcomes. Additionally, we address selection bias through our orthogonal diffusion loss.

In the denoising matching term, we try to minimize the KL-divergence between two distributions, one is tractable, ground truth denoising transition step $q\left(y_{t-1} \mid y_t, y_0\right)$, and other is denoising transition step $p_\theta\left(y_{t-1} \mid y_t, x, a\right)$.

**Conditional denoising function.** In practice, directly optimizing the learning objective in Eq. (7) is inefficient. Recently, [26] has shown a way to turn the optimization of the ELBO into a simpler problem for *unconditional* diffusion models. We thus follow a similar way to obtain a simpler problem for our learning objective, but for our *conditional* diffusion models.

We employ a trainable conditional denoising function $f_\theta : (\mathcal{Y} \times \mathbb{R} \mid \mathcal{X}, \mathcal{A}) \to \mathcal{Y}$. The conditional denoising function is a function approximation, which we use to predict the 'ground truth' noise component $\epsilon$ for the noisy data sample $y_t$ conditioned on $x$ and $a$. We consider the following parameterization that computes $\mu_\theta\left(y_t, t \mid x, a\right)$ as a function of $y_t, x, a$ and $f_\theta$. This gives

$$\mu_\theta\left(y_t, t \mid x, a\right) = \frac{1}{\sqrt{\alpha_t}}\left(y_t - \frac{\beta_t}{\sqrt{1 - \bar{\alpha}_t}} f_\theta\left(y_t, t \mid x, a\right)\right). \tag{8}$$

The simplified training objective for $t \sim \mathrm{Unif}\{1, \ldots, T\}$ is then

$$\mathbb{E}_{(y_0, x, a) \sim p(Y, X, A); \epsilon \sim \mathcal{N}(\mathbf{0}, \mathbf{I})}\left[\left\|\epsilon - f_\theta\left(\sqrt{\bar{\alpha}_t} y_0 + \sqrt{1 - \bar{\alpha}_t} \epsilon, t \mid x, a\right)\right\|^2\right]. \tag{9}$$

We provide the full derivation of Eq. (8) and Eq. (9) in Appendix C.

## 5.2 Orthogonal diffusion loss for addressing selection bias

*Why we need to address selection bias through our orthogonal diffusion loss:* In observational data, treatments are not randomized but are administered according to some (unknown) behavioral policy. This can result in a *selection bias*, especially in medical practice. For example, patients with a more severe health state will also receive a more aggressive treatment. As a result, when treatment $A$ is selected based on the covariates $X$, the propensity score $\pi(X)$ is not constant, implying that the distributions of covariates in treatment and control groups differ. Formally, we have a distribution shift in the covariates, i.e., $p(X \mid A = 0) \neq p(X \mid A = 1)$. If not adjusted for, this distribution shift may result in a *selection bias* and thus an *inflated variance* in the estimates of POs (and thus the causal effects), especially in low-sample settings [12, 29, 54]. To address this, we thus introduce a novel orthogonal diffusion in the following.

③ **Orthogonal diffusion loss.** Our proposed orthogonal diffusion loss is inspired by orthogonal learning theory [18, 60]. Orthogonal learning theory provides a general toolbox for inference with semi-parametric models to provide quasi-oracle rates for statistical learning with a nuisance component. Informally, orthogonal losses are first-order insensitive to misspecification of the nuisance functions, which introduces many favorable properties such as double robustness [63]. Below, we construct an orthogonal diffusion loss for our method.

We start by noting that $\pi(x)$ is not constant and is unknown in observational datasets. Because of that, one cannot simply use the propensity scores to reweight the data. Instead, we need a trainable approach to reweight the data and thus adjust for the distribution shift from above. To this end, the propensity score $\pi(x)$ is a nuisance function in our method, which we estimate through a trainable

function $g_\phi$ parameterized by $\phi$. Let us denote the estimated propensity score by $\hat\pi(x) = g_\phi(x)$. We then assign weights $w_{\hat\pi}(x, a)$ to each sample via some function $w_{\hat\pi} : (\mathcal{X} \times \{0, 1\}) \to \mathbb{R}_+$. More specifically, we define the function $w_{\hat\pi} : (\mathcal{X} \times \{0, 1\}) \to \mathbb{R}_+$ as

$$w_{\hat\pi}(x, a) = \frac{a}{\hat\pi(x)} + \frac{1 - a}{1 - \hat\pi(x)}. \tag{10}$$

Then, our *orthogonal diffusion loss* is given by

$$\mathcal{L}(\theta, \hat\pi) = \mathbb{E}_{(y_0, x, a) \sim p(Y, X, A); \epsilon \sim \mathcal{N}(\mathbf{0}, \mathbf{I})} \left[ w_{\hat\pi}(x, a) \left\| \epsilon - f_\theta \left( \sqrt{\bar\alpha_t} y_0 + \sqrt{1 - \bar\alpha_t} \epsilon, t \mid x, a \right) \right\|^2 \right]. \tag{11}$$

The loss in Eq. (9) fits the conditional distributions $p(Y \mid X, A)$ based on the data samples from the joint observational distribution, i.e., $(y_0, x, a) \sim p(Y, X, A)$. Due to the selection bias, this implies that $p(Y \mid X, A = 1)$ is learned better for the treated population and $p(Y \mid X, A = 0)$ for the untreated population [60]. Yet, our target is to learn the potential outcome distributions $p(Y(a) \mid X) = p(Y \mid X, A = a)$ for both $a = 0$ and $a = 1$ equally well in all the populations. This would be equivalent to minimizing the target loss from Eq. (9) if the samples are from the joint distributions of the POs, i.e., $(y_0, x) \sim p(Y(a), X)$. However, we do not have samples from the distributions of POs, thus we need the target loss in Eq. (11) to be estimated via the propensity score $\pi(x)$ to learn the distributions of POs. Hence, the following remark holds.

**Remark 1.** *The orthogonal diffusion loss in Eq. (11) evaluated with the ground truth nuisance functions matches the following target loss:*

$$\mathcal{L}(\theta, \pi) = \sum_{a \in \{0, 1\}} \mathbb{E}_{(y_0, x) \sim p(Y(a), X); \epsilon \sim \mathcal{N}(\mathbf{0}, \mathbf{I})} \left[ \left\| \epsilon - f_\theta \left( \sqrt{\bar\alpha_t} y_0 + \sqrt{1 - \bar\alpha_t} \epsilon, t \mid x, a \right) \right\|^2 \right]. \tag{12}$$

*Proof.* The orthogonal diffusion loss in Eq. (11) is an inverse propensity-weighted estimator of the target loss in Eq. (12). □

**Theorem 1** (Neyman-orthogonality). *The orthogonal diffusion loss in Eq. (11) is Neyman-orthogonal wrt. its nuisance functions.*

*Proof.* See Appendix D. □

As a result of Neyman-orthogonality, our orthogonal diffusion loss has a clear practical advantage: it offers robustness against errors in nuisance function estimation. Specifically, it is first-order insensitive wrt. errors in the propensity score $\pi(x)$ and, thus, is a more accurate objective for our PO predictive model.

## 5.3 Training and sampling

**Training.** Our training algorithm proceeds along the following steps. We first train the function $g_\phi$ on the dataset to estimate the propensity score. After this, the parameters of $g_\phi$ are frozen. We then compute each sample weight for the orthogonal diffusion loss through the weight function $w_{\hat\pi}$. We finally sample from the observational data distribution and run the forward and reverse process as described in Sec. 5.1. The corresponding training loss is given in Eq. (11).

**Sampling.** Once our diffusion model has learned the distribution $p_\theta(Y_{t-1} \mid y_t, x, a)$, we can generate samples from it. The sampling process starts by sampling $y_T$ from the prior distribution and then denoising it. To obtain $y_{t-1}$ from $p_\theta(y_{t-1} \mid y_t, x, a)$, this requires the mean $\tilde\mu$ from the previous step, which can be computed by Eq. (8). Thus, $y_{t-1}$ can be computed via $\tilde\mu + \sigma_t z$, where $z \sim \mathcal{N}(\mathbf{0}, \mathbf{I})$. This process continues for $t = T, \ldots$ until we arrive at $y_0$.

**Implementation.** Our model architecture of denoising function is based on [59] and uses U-Net [49] as the underlying backbone. We use 4 residual blocks where each is built with MLP layers. The diffusion embedding dimension is 128. The number of diffusion sampling steps is 100. Training is conducted with a batch size of 256 and a learning rate of 0.0005. We follow [42] in the way how we learn propensity scores and thus use fully connected neural networks with softmax activation. During training, we can only observe one of the two POs due to the fundamental problem of causal inference [27] (as explained in Sec. 4). To guide the model, we need to identify which PO should the loss be

computed on. Hence, we introduce causal masks as input to our model: an observational mask $\mathrm{m}_o$, a targeted mask $\mathrm{m}_t$, and a conditional mask $\mathrm{m}_c$. We only compute the loss at the place where the value of the targeted mask is 1. Further details about our implementation are in Appendix E.1.

## 6 Experiments

### 6.1 Learning distributions of POs

**Performance metrics.** We use the Wasserstein distance to evaluate the model performance of learning the distributions of POs. The $k$-Wasserstein distance (for any $k \geq 1$) between two distributions $\nu_1$ and $\nu_2$ is

$$W^k(\nu_1, \nu_2) = \left( \int_0^1 \left| \mathbb{F}_1^{-1}(l) - \mathbb{F}_2^{-1}(l) \right|^k \, \mathrm{d}l \right)^{1/k}, \tag{13}$$

where $\mathbb{F}_1^{-1}(l)$ and $\mathbb{F}_2^{-1}(l)$ are the quantile functions (inverse cumulative distribution functions) of $\nu_1$ and $\nu_2$ for quantile $l$, respectively. Specifically, we compute the empirical Wasserstein distance based on two sets of finite samples, i.e., $W^k(\nu_1, \nu_2) = \left( \frac{1}{n} \sum_{i=1}^n \left\| X_{(i)} - Y_{(i)} \right\|^k \right)^{1/k}$, where $X_1, \ldots, X_n$ are the samples from $\nu_1$ and $Y_1, \ldots, Y_n$ are the samples from $\nu_2$. In our experiments, we report the empirical Wasserstein distance for $k = 1$. Lower values of $W^k$ are preferred.

**Dataset.** Due to the fundamental problem of causal inference, the counterfactual outcomes are never observed in real-world data. We thus follow prior literature (e.g.,[22, 38]) and benchmark our model using synthetic datasets. Further details about the synthetic datasets are in the Appendix F.

**Baselines.** Many methods are targeting CATE estimation and, therefore, are *not* suitable for PO prediction. We thus compare against those methods that are applicable to our task (i.e., the model can directly output POs; see Table 1): (1) **S-learner** [37]: is a model-agnostic learner that fits a single regression model by concatenat-

Table 2: Results showing in- & out-of-sample empirical Wasserstein distance (i.e., $\hat{W}_{\mathrm{in}}^1$ and $\hat{W}_{\mathrm{out}}^1$) for two potential outcomes (i.e., $a = 0$ and $a = 1$) on the synthetic dataset. Reported: mean $\pm$ standard deviation over ten-fold train-test splits.

| | $a = 0$ | | $a = 1$ | |
|---|---|---|---|---|
| | $\hat{W}_{\mathrm{in}}^1$ | $\hat{W}_{\mathrm{out}}^1$ | $\hat{W}_{\mathrm{in}}^1$ | $\hat{W}_{\mathrm{out}}^1$ |
| S-learner* [37] | $1.007 \pm 0.121$ | $1.005 \pm 0.234$ | $1.781 \pm 0.092$ | $1.955 \pm 0.233$ |
| T-learner* [37] | $1.034 \pm 0.126$ | $1.002 \pm 0.298$ | $1.054 \pm 0.092$ | $1.453 \pm 0.172$ |
| TARNet* [54] | $0.934 \pm 0.155$ | $0.978 \pm 0.167$ | $0.922 \pm 0.183$ | $1.211 \pm 0.212$ |
| CFR* [54] | $0.921 \pm 0.112$ | $0.943 \pm 0.145$ | $0.909 \pm 0.112$ | $1.078 \pm 0.192$ |
| GANITE* [64] | $0.759 \pm 0.325$ | $1.324 \pm 0.229$ | $0.821 \pm 0.343$ | $1.112 \pm 0.348$ |
| **DiffPO (ours)** | $\mathbf{0.043 \pm 0.021}$ | $\mathbf{0.125 \pm 0.051}$ | $\mathbf{0.032 \pm 0.037}$ | $\mathbf{0.091 \pm 0.055}$ |

Lower = better (best in bold). * modified method to make it comparable.

ing the covariate and the treatment as input; (2) **T-learner** [37]: is a model-agnostic learner that fits two separate regression models, one for treated and for controls; (3) **TARNet** [54]: is based on representation learning to share information about the outcome across treated and controls with regularization; (4) **CFR** [54]: is based on representation learning used in variants of balancing with TARNet; (5) **GANITE** [64]: uses a generative adversarial network to generate POs and then uses another generative adversarial network to generate CATE. We adapt GANITE to our task by removing the second stage, so that we directly sample POs from the learned distributions in the first stage. However, methods (1) to (4) above are only able to give point estimates of POs. Therefore, we follow [24] to equip them with Monte Carlo (MC) dropout to make these methods comparable. Implementation details are in Appendix E.2.

**Results.** The results are in Table 2. We find that our method gives the lowest empirical Wasserstein distance out of all methods, which is desirable. Hence, the experiments show that our method outperforms the baselines by a clear margin.

### 6.2 Learning predictive intervals of POs

**Performance metrics.** To evaluate the ability of our DiffPO to allow for uncertainty estimation, we follow [24] and examine the predictive intervals (PIs) of the POs generated by different methods. In medical practice, the treatment effectiveness is often reported based on the 95% and 99% PIs. We thus compute the individual equal-tailed $(1 - \alpha)$ PIs with $\alpha \in [0.01, 0.05]$. We then calculate the *faithfulness* of the estimated PIs to evaluate the empirical coverage by computing the frequency with which the PIs contain the outcomes in the test data.

**Results:** We evaluate the empirical coverage across different quantiles $(1 - \alpha)$. The results are in Table 3. It shows that our DiffPO is generally faithful, while this is not the case for the baselines. This can be expected: MC dropout relies upon mixtures of Dirac distributions in parameter space, which leads to approximations of the true posterior which are questionable and not faithful [17, 24].

### 6.3 Point estimates of POs

**Performance metrics for POs:** To evaluate point estimates of POs, we compute the difference between the predicted PO $\hat{y}_i$ and the ground truth PO $y_i$ via the root mean squared error RMSE $= \sqrt{\frac{1}{N} \sum_{i=1}^{N} (\hat{y}_i - y_i)^2}$.

**Baselines.** We compare against those methods that are applicable to this task (i.e., the model can directly output POs; see Table 1).

**Results.** The results are in Table 4. We find that our DiffPO gives the best point estimates of POs.

Table 3: Results for uncertainty estimation of the two potential outcomes (i.e., $a = 0$ and $a = 1$). Reported: mean $\pm$ standard deviation over ten-fold train-test splits.

| | $a = 0$ | | $a = 1$ | |
|---|---|---|---|---|
| | 95% PI | 99% PI | 95% PI | 99% PI |
| S-learner* [37] | 0.801 $_{\pm 0.05}$ | 0.891 $_{\pm 0.05}$ | 0.879 $_{\pm 0.05}$ | 0.898 $_{\pm 0.05}$ |
| T-learner* [37] | 0.832 $_{\pm 0.05}$ | 0.854 $_{\pm 0.05}$ | 0.843 $_{\pm 0.05}$ | 0.821 $_{\pm 0.05}$ |
| TARNet* [54] | 0.855 $_{\pm 0.08}$ | 0.864 $_{\pm 0.08}$ | 0.845 $_{\pm 0.08}$ | 0.878 $_{\pm 0.08}$ |
| CFR* [54] | 0.861 $_{\pm 0.08}$ | 0.954 $_{\pm 0.08}$ | 0.853 $_{\pm 0.08}$ | 0.862 $_{\pm 0.08}$ |
| GANITE* [64] | 0.892 $_{\pm 0.16}$ | 0.902 $_{\pm 0.16}$ | 0.873 $_{\pm 0.16}$ | 0.885 $_{\pm 0.16}$ |
| **DiffPO** (Ours) | **0.981** $_{\pm 0.02}$ | **0.985** $_{\pm 0.01}$ | **0.908** $_{\pm 0.01}$ | **0.926** $_{\pm 0.02}$ |

Higher = better (best in bold). * modified for comparability.

Table 4: Results for point estimation of POs benchmarked using the in- & out-of-sample RMSE for the two potential outcomes (i.e., $a = 0$ and $a = 1$) on the synthetic dataset. Reported: mean $\pm$ standard deviation over ten-fold train-test splits.

| | $a = 0$ | | $a = 1$ | |
|---|---|---|---|---|
| | RMSE$_{in}$ | RMSE$_{out}$ | RMSE$_{in}$ | RMSE$_{out}$ |
| S-learner [37] | 0.401 $_{\pm 0.05}$ | 0.391 $_{\pm 0.05}$ | 0.379 $_{\pm 0.05}$ | 0.398 $_{\pm 0.05}$ |
| T-learner [37] | 0.432 $_{\pm 0.05}$ | 0.454 $_{\pm 0.05}$ | 0.443 $_{\pm 0.05}$ | 0.421 $_{\pm 0.05}$ |
| TARNet [54] | 0.364 $_{\pm 0.06}$ | 0.388 $_{\pm 0.06}$ | 0.319 $_{\pm 0.06}$ | 0.382 $_{\pm 0.06}$ |
| CFR [54] | 0.263 $_{\pm 0.06}$ | 0.278 $_{\pm 0.06}$ | 0.261 $_{\pm 0.06}$ | 0.293 $_{\pm 0.06}$ |
| GANITE [64] | 0.378 $_{\pm 0.16}$ | 0.390 $_{\pm 0.16}$ | 0.525 $_{\pm 0.16}$ | 0.547 $_{\pm 0.16}$ |
| **DiffPO** (Ours) | **0.143** $_{\pm 0.14}$ | **0.156** $_{\pm 0.14}$ | **0.162** $_{\pm 0.14}$ | **0.187** $_{\pm 0.14}$ |

Lower = better (best in bold).

### 6.4 Flexibility to handle other causal quantities

A strength of our method is that it is flexible and can not only be used for POs but also for other causal quantities. For example, even though we are aiming at learning the distributions of POs, our method is also capable of estimating the CATE $\tau(x) = \mathbb{E}[Y(1) - Y(0) \mid X = x]$, which is the expected difference of POs for an individual with covariate values $X = x$. For this, one can use our method to first predict $\mu_a(x) = \mathbb{E}[Y \mid X, A = a]$ for $a \in \{0, 1\}$ and then simply leverage that $\tau(x) = \mu_1(x) - \mu_0(x)$.

**Datasets.** We estimate the CATE across ACIC 2016 & ACIC 2018, which are widely used dataset collections for CATE benchmarking [66, 12, 42]. The ACIC2016 [46] contains 77 different benchmark datasets, and ACIC2018 [41] contains 24. These datasets include a wide range of data-generating mechanisms (see Appendix F for more details). We use five random train/test splits (80% / 20%) for each dataset, tune hyperparameters on the first split, and evaluate the average out-sample on every split.

**Performance metrics.** We compare the $\epsilon_{\text{PEHE}} = \sqrt{\frac{1}{N} \sum_{i=1}^{N} (\hat{\tau}(x_i) - \tau(x_i))^2}$.

Table 5: Results for benchmarking CATE estimation on ACIC 2016 and ACIC 2018 for both in- & out-of-sample, respectively. Reported: % of runs with the best performance.

| | ACIC 2016 (77 datasets) | | ACIC 2018 (24 datasets) | |
|---|---|---|---|---|
| | % best$_{in}$ | % best$_{out}$ | % best$_{in}$ | % best$_{out}$ |
| S-learner [37] | 3.76 | 3.41 | 1.16 | 1.02 |
| T-learner [37] | 3.71 | 3.8 | 0.74 | 1.13 |
| TARNet [54] | 8.92 | 9.21 | 7.98 | 6.71 |
| CFR [54] | 10.52 | 11.45 | 7.91 | 7.71 |
| GANITE [64] | 6.78 | 6.44 | 5.22 | 3.98 |
| DR-learner [19] | 14.54 | 15.67 | 23.64 | 22.13 |
| RA-learner [12] | 15.71 | 14.68 | 15.73 | 15.01 |
| TEDVAE [66] | 9.33 | 9.52 | 7.28 | 5.45 |
| **DiffPO** | **26.73** | **25.82** | **31.34** | **36.86** |

Higher = better (best in bold).

**Baselines.** We consider a broad array of state-of-the-art methods for treatment effect estimation from the literature. Considering the rich methods in the literature, we thus compare with the most popular CATE estimation approaches in this field [10, 30, 37, 54, 61, 64, 66], as listed in the Table 1. Specifically, we use the following baselines: **(1)** to **(5)** from Sec. 6.1. (6) **DR-learner** [39, 31]: generates pseudo-outcomes based on the doubly-robust AIPW estimator; (7) **RA-learner** [12]: uses a regression-adjusted pseudo-outcome in the second stage; (8) **TEDVAE** [66]: uses a variational autoencoder to differentiate confounding factor and learns CATE. We instantiate the meta-learners (i.e., S-, T-, DR-, and RA-learner) with neural networks, similar to our DiffPO. To ensure a fair comparison across all methods and all experiments, we tune hyperparameters across all methods

separately for each experimental dataset. We discuss implementation details of baselines in the Appendix E.

**Results.** The results for CATE estimation are in Table 5. We find that our method achieves a good performance similar to existing methods, even though our method is not tailored for CATE estimation. Across a wide range of experiments in different settings, we even observe that our method often achieves state-of-the-art performance.

## 6.5 Visualizing the learned distributions of POs

A clear advantage of our DiffPO is that we not only return point estimates but also the *distribution* of POs. This is crucial in medical practice [23, 35] in order to understand the expected probability of whether a treatment is beneficial and thus to better assess the reliability of the estimates. We show the learned distributions for a real-world dataset from medicine.

**IHDP dataset.** This is a semi-synthetic dataset from the Infant Health and Development Program (IHDP) [25], which is based on the extracted features and treatment assignments from a real-world clinical trial. The dataset comprises 747 patients, with 25 features for each patient. Further details for IHDP are in Appendix F.3.

**Insights.** Our DiffPO is capable of capturing the full information about the distributions of POs. Current state-of-the-art methods usually estimate quantities expressed via the mean of POs, as these methods focus purely on point estimation [31, 37, 54]. However, distributional knowledge of POs is important to account for uncertainty, because it informs how likely a certain outcome is and gives the probability that the PO lies in a desired range, especially in medicine.

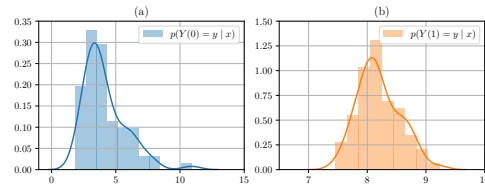

Figure 2: Empirical distributions of the conditional POs. Left: $p(Y(0) \mid x)$. Right: $p(Y(1) \mid x)$.

Fig. 2 shows the empirical distributions of POs, i.e., $p(Y(0) \mid X)$ and $p(Y(1) \mid X)$, given a certain patient profile $X = x$. We can see the distributions of the two POs are different, which can provide extra information for medical decision-making. Here, the learned distributions of POs can help medical practitioners in choosing a treatment plan that promises not only a large benefit for the patients but also that the benefit is highly probable.

# 7  Discussion

**Conclusion.** Our method is carefully designed for predicting POs in medical practice. To this end, our method not only predicts point estimates but also computes *the distributions of POs to allow for uncertainty quantification and thus for reliable decision-making*.

**Limitations.** (1) As with other methods in causal inference, ours rests on mathematical assumptions, yet these are standard in the literature [12, 31, 64]. (2) The efficiency of the sampling process in our DiffPO – but also diffusion models more generally – could be improved further. There is already ongoing research, such as different solvers [15, 40, 65] and one-step sampling [58]. However, as we have shown above, our method can successfully scale to real-world datasets from medical practice.

**Broader impact.** We expect our method to have a significant impact in medicine where better decision support is needed to personalize treatment decisions to patient profiles [16]. Another strength of our method is that it is flexible. This is unlike many other methods in causal inference which are designed for highly restrictive settings. Hence, we expect our method to be a first step toward developing more generalizable approaches for a variety of causal inference settings.

**Acknowledgements**

This paper is supported by the DAAD program "Konrad Zuse Schools of Excellence in Artificial Intelligence", sponsored by the Federal Ministry of Education and Research. This work has been supported by the German Federal Ministry of Education and Research (Grant: 01IS24082).

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

# A Extended related work

## A.1 Diffusion models for causal inference

Diffusion models were originally introduced for learning a complex distribution for a given dataset from which we can then sample high-quality images [55, 57, 26]. Diffusion models have achieved state-of-the-art performance, outperforming other generative models on various tasks in the computer vision field (e.g.,[56, 14, 59]). Motivated by this, diffusion models were previously used for different causal inference tasks but in a different setting from ours [51, 52, 8].

[51] proposed a model called Diff-SCM that uses diffusion models and anti-causal predictors to generate the counterfactual of a given image. It leverages a classifier guidance diffusion model. [8] proposed using a diffusion model called DCM for modeling structural causal models (SCMs). It focuses on approximating SCMs given observational data and underlying causal DAG. [8] focused on a similar task as well as [53, 36], with the aim to answer causal queries by employing variational autoencoder and normalizing flow, respectively. [36] focused on the same task as [51], aiming for generating high-quality counterfactual images and proposed a model called CausalDiffAE. [36] follows the idea of [8] to view the diffusion model as an encoder-decoder framework and employ so-called denoising diffusion implicit models (DDIM). Diffusion models have also been used in the causal discovery field. DiffAN [52] is an algorithm based on denoising diffusion training for causal discovery via topological ordering. In this work, we use diffusion models for predicting POs.

## A.2 Neyman-orthogonality

Neyman-orthogonality of a functional refers to the mean zero property of its directional derivatives along one-dimensional paths that change nuisance functions [18]. A loss function is called Neyman-orthogonal when this property holds for all its directional derivatives along one-dimensional paths that change the (infinite-dimensional) parameter of interest. This allows for a debiased machine learning approach to construct estimators and inference procedures that are robust to small mistakes in nuisance functions [9]. Specific orthogonal learners for estimating CATE are given in the DR-learner [39, 31], R-learner [45], and i-learner [60].

## A.3 Conditional average treatment effect (CATE)

Our work is about predicting potential outcomes (POs) for individuals; however, it is strongly related in practical applications to conditional average treatment effect (CATE) estimation. The PO framework conceptualizes the CATE estimation problem as estimating the expected difference between an individual's expected potential outcome with and without treatment. Below, we give a brief overview of the CATE literature.

The estimation of the CATE has received a lot of attention in the causal inference literature (e.g.,[10, 25, 28, 1, 54, 37, 2, 30, 3, 64, 61, 12, 66, 31, 45, 32, 13]). Among those CATE estimation methods, many have been proposed to estimate the effects of binary treatments (e.g.,[10, 25, 28, 1, 54, 37, 2, 30, 3, 64, 6, 12, 66, 31, 45, 32, 13]). Based on the categorization in [13], we roughly categorize them into three categories based on their most salient characteristics: (i) model-agnostic learning strategies for CATE estimation (also known as meta-learners), and (ii) model-specific ML-based CATE estimators (including representation learning-based and other neural network-based estimators).

### A.3.1 Meta-learners

Now we consider popular approaches that involve using model-agnostic learning strategies, also known as meta-learners, which can be implemented using any ML method (e.g.,[37, 31, 45, 32, 13, 12]). The idea behind "meta-learner" strategies was originally introduced in [37] and expanded in [45, 31, 12].

Existing CATE meta-learners can be categorized into: (a) one-step (plug-in) learners (indirect meta-learners) that output two regression functions from the observational data and then compute CATE as the difference in the POs (this is the strategy underlying the S- and T-learners); and (b) two-step learners (direct meta-learners/multi-stage direct estimators). These learners first compute nuisance functions to build a pseudo-outcome. In the second step, they obtain the CATE directly by regressing the input covariates on the pseudo-outcome. (Note that *pseudo-outcomes are not potential outcomes*).

In terms of (b), existing methods fall largely into three broad classes: regression adjustment (RA), propensity weighting (PW), or doubly robust (DR) strategies.

### A.3.2  Model-specific ML-based CATE estimators

Many methods proposed in the related work do not fall within the meta-learner class because they rely on the properties of a specific ML method. We roughly categorize them into two branches: one is neural network-based (NN-based) estimators and the other does not use neural networks, for example, some using causal Forest [61], Bayesian regression [25], or Gaussian processes [1].

As for the NN-based estimators, some are representation learning-based estimators. Much work of this track focused on handling selection bias by learning shared and balanced feature representations for the two PO functions or incorporating weighting strategies. Examples are BNN [28], TARNet [54], and CFR [30]. [54] used representation learning to share information about the outcome across treated and controls, while regularizing representation distributional distance between the groups. [67, 30] recognized that such regularization may result in a violation of ignorability in the representation and proposed to learn representations in which context information is preserved but where treatment groups overlap. Later, many works proposed further extensions (e.g.,[29, 20, 4, 21, 62, 67, 5, 7]). However, a wrongly chosen dimensionality of the representation or a too large balancing weight can induce confounding bias [43], and, therefore, the representation learning-based methods fall outside the scope of this paper.

# B  Evidence lower bound of DiffPO

Directly computing and maximizing the likelihood $p(y \mid x, a)$ is difficult because it involves integrating all latent variables, which is intractable for complex models. Thus, we derive the evidence lower bound (ELBO) similar to the bound of the variational autoencoder. Our original objective can be optimized by maximizing the ELBO, like its analog in the ELBO of a vanilla VAE [34], which is given by

$$
\begin{aligned}
\log p(y \mid x, a) &= \log \int p\left(y_{0:T} \mid x, a\right) dy_{1:T} \\
&= \log \int \frac{p\left(y_{0:T} \mid x, a\right) q\left(y_{1:T} \mid y_0\right)}{q\left(y_{1:T} \mid y_0\right)} dy_{1:T} \\
&= \log \mathbb{E}_{q(y_{1:T}|y_0)} \left[ \frac{p\left(y_{0:T} \mid x, a\right)}{q\left(y_{1:T} \mid y_0\right)} \right] \\
&\geq \mathbb{E}_{q(y_{1:T}|y_0)} \left[ \log \frac{p\left(y_{0:T} \mid x, a\right)}{q\left(y_{1:T} \mid y_0\right)} \right].
\end{aligned} \tag{14}
$$

The ELBO in Eq. (14) can be rewritten as follows:

$$
\begin{aligned}
\log p(y) \geq & \mathbb{E}_{q(y_{1:T}|y_0)} \left[ \log \frac{p(y_{0:T} \mid x, a)}{q(y_{1:T} \mid y_0)} \right] \\
= & \mathbb{E}_{q(y_{1:T}|y_0)} \left[ \log \frac{p(y_T) \prod_{t=1}^{T} p_\theta(y_{t-1} \mid y_t, x, a)}{\prod_{t=1}^{T} q(y_t \mid y_{t-1})} \right] \\
= & \mathbb{E}_{q(y_{1:T}|y_0)} \left[ \log \frac{p(y_T) p_\theta(y_0 \mid y_1, x, a) \prod_{t=2}^{T} p_\theta(y_{t-1} \mid y_t, x, a)}{q(y_1 \mid y_0) \prod_{t=2}^{T} q(y_t \mid y_{t-1})} \right] \\
= & \mathbb{E}_{q(y_{1:T}|y_0)} \left[ \log \frac{p(y_T) p_\theta(y_0 \mid y_1, x, a) \prod_{t=2}^{T} p_\theta(y_{t-1} \mid y_t, x, a)}{q(y_1 \mid y_0) \prod_{t=2}^{T} q(y_t \mid y_{t-1}, y_0)} \right] \\
= & \mathbb{E}_{q(y_{1:T}|y_0)} \left[ \log \frac{p(y_T) p_\theta(y_0 \mid y_1, x, a)}{q(y_1 \mid y_0)} + \log \prod_{t=2}^{T} \frac{p_\theta(y_{t-1} \mid y_t, x, a)}{q(y_t \mid y_{t-1}, y_0)} \right] \\
= & \mathbb{E}_{q(y_{1:T}|y_0)} \left[ \log \frac{p(y_T) p_\theta(y_0 \mid y_1, x, a)}{q(y_1 \mid y_0)} + \log \prod_{t=2}^{T} \frac{p_\theta(y_{t-1} \mid y_t, x, a)}{\frac{q(y_{t-1}|y_t, y_0) q(y_t|y_0)}{q(y_{t-1}|y_0)}} \right] \\
= & \mathbb{E}_{q(y_{1:T}|y_0)} \left[ \log \frac{p(y_T) p_\theta(y_0 \mid y_1, x, a)}{q(y_1 \mid y_0)} + \log \frac{q(y_1 \mid y_0)}{q(y_T \mid y_0)} + \log \prod_{t=2}^{T} \frac{p_\theta(y_{t-1} \mid y_t, x, a)}{q(y_{t-1} \mid y_t, y_0)} \right] \\
= & \mathbb{E}_{q(y_{1:T}|y_0)} \left[ \log \frac{p(y_T) p_\theta(y_0 \mid y_1, x, a)}{q(y_T \mid y_0)} + \sum_{t=2}^{T} \log \frac{p_\theta(y_{t-1} \mid y_t, x, a)}{q(y_{t-1} \mid y_t, y_0)} \right] \\
= & \mathbb{E}_{q(y_{1:T}|y_0)} \left[ \log p_\theta(y_0 \mid y_1, x, a) \right] + \\
& \quad \mathbb{E}_{q(y_{1:T}|y_0)} \left[ \log \frac{p(y_T)}{q(y_T \mid y_0)} \right] + \sum_{t=2}^{T} \mathbb{E}_{q(y_{1:T}|y_0)} \left[ \log \frac{p_\theta(y_{t-1} \mid y_t, x, a)}{q(y_{t-1} \mid y_t, y_0)} \right] \\
= & \mathbb{E}_{q(y_1|y_0)} \left[ \log p_\theta(y_0 \mid y_1, x, a) \right] + \\
& \quad \mathbb{E}_{q(y_T|y_0)} \left[ \log \frac{p(y_T)}{q(y_T \mid y_0)} \right] + \sum_{t=2}^{T} \mathbb{E}_{q(y_t, y_{t-1}|y_0)} \left[ \log \frac{p_\theta(y_{t-1} \mid y_t, x, a)}{q(y_{t-1} \mid y_t, y_0)} \right] \\
= & \underbrace{\mathbb{E}_{q(y_1|y_0)} \left[ \log p_\theta(y_0 \mid y_1, x, a) \right]}_{\text{reconstruction term}} - \underbrace{D_{\mathrm{KL}} \left( q(y_T \mid y_0) \| p(y_T) \right)}_{\text{prior matching term}} \\
& - \sum_{t=2}^{T} \underbrace{\mathbb{E}_{q(y_t|y_0)} \left[ D_{\mathrm{KL}} \left( q(y_{t-1} \mid y_t, y_0) \| p_\theta(y_{t-1} \mid y_t, x, a) \right) \right]}_{\text{denoising matching term}}.
\end{aligned}
$$

$$(15)$$

## C  Simplified training objective for DiffPO

With the reverse process defined above, the learning objective in Eq. (7) is clearly differentiable with respect to $\theta$ and is ready to be employed for training. However, in practice, directly optimizing this objective is inefficient. Recall that our aim is to match the approximate denoising transition step $p_\theta(y_{t-1} \mid y_t, x, a)$ to ground truth denoising transition step $q(y_{t-1} \mid y_t, y_0)$ as closely as possible, which we can also model as a Gaussian. Thus, optimizing the KL divergence term reduces to minimizing the difference between the means of the two distributions:

$$
\begin{aligned}
&\arg\min_\theta D_{\mathrm{KL}}\left(q\left(y_{t-1} \mid y_t, y_0\right) \,\|\, p_\theta\left(y_{t-1} \mid y_t, x, a\right)\right) \\
=&\arg\min_\theta D_{\mathrm{KL}}\left(\mathcal{N}\left(y_{t-1}; \mu_q, \Sigma_q(t)\right) \,\|\, \mathcal{N}\left(y_{t-1}; \mu_\theta, \Sigma_q(t)\right)\right) \\
=&\arg\min_\theta \frac{1}{2}\left[\log\frac{|\Sigma_q(t)|}{|\Sigma_q(t)|} - d + \mathrm{tr}\left(\Sigma_q(t)^{-1}\Sigma_q(t)\right) + \left(\mu_\theta - \mu_q\right)^T \Sigma_q(t)^{-1}\left(\mu_\theta - \mu_q\right)\right] \\
=&\arg\min_\theta \frac{1}{2}\left[\log 1 - d + d + \left(\mu_\theta - \mu_q\right)^T \Sigma_q(t)^{-1}\left(\mu_\theta - \mu_q\right)\right] \\
=&\arg\min_\theta \frac{1}{2}\left[\left(\mu_\theta - \mu_q\right)^T \Sigma_q(t)^{-1}\left(\mu_\theta - \mu_q\right)\right] \\
=&\arg\min_\theta \frac{1}{2}\left[\left(\mu_\theta - \mu_q\right)^T \left(\sigma_q^2(t)\mathbf{I}\right)^{-1}\left(\mu_\theta - \mu_q\right)\right] \\
=&\arg\min_\theta \frac{1}{2\sigma_q^2(t)}\left[\|\mu_\theta - \mu_q\|_2^2\right].
\end{aligned}
\tag{16}
$$

By using the reparameterization trick, we have

$$
y_0 = \frac{y_t - \sqrt{1 - \bar{\alpha}_t}\epsilon_0}{\sqrt{\bar{\alpha}_t}}.
\tag{17}
$$

Plugging it into the previously derived true denoising transition mean $\mu_q(y_t, y_0)$, we yield

$$
\begin{aligned}
\mu_q\left(y_t, y_0\right) &= \frac{\sqrt{\alpha_t}\left(1 - \bar{\alpha}_{t-1}\right)y_t + \sqrt{\bar{\alpha}_{t-1}}\left(1 - \alpha_t\right)y_0}{1 - \bar{\alpha}_t} \\
&= \frac{\sqrt{\alpha_t}\left(1 - \bar{\alpha}_{t-1}\right)y_t + \sqrt{\bar{\alpha}_{t-1}}\left(1 - \alpha_t\right)\frac{y_t - \sqrt{1-\bar{\alpha}_t}\epsilon_0}{\sqrt{\bar{\alpha}_t}}}{1 - \bar{\alpha}_t} \\
&= \frac{\sqrt{\alpha_t}\left(1 - \bar{\alpha}_{t-1}\right)y_t + \left(1 - \alpha_t\right)\frac{y_t - \sqrt{1-\bar{\alpha}_t}\epsilon_0}{\sqrt{\alpha_t}}}{1 - \bar{\alpha}_t} \\
&= \frac{\sqrt{\alpha_t}\left(1 - \bar{\alpha}_{t-1}\right)y_t}{1 - \bar{\alpha}_t} + \frac{\left(1 - \alpha_t\right)y_t}{\left(1 - \bar{\alpha}_t\right)\sqrt{\alpha_t}} - \frac{\left(1 - \alpha_t\right)\sqrt{1 - \bar{\alpha}_t}\epsilon_0}{\left(1 - \bar{\alpha}_t\right)\sqrt{\alpha_t}} \\
&= \left(\frac{\sqrt{\alpha_t}\left(1 - \bar{\alpha}_{t-1}\right)}{1 - \bar{\alpha}_t} + \frac{1 - \alpha_t}{\left(1 - \bar{\alpha}_t\right)\sqrt{\alpha_t}}\right)y_t - \frac{\left(1 - \alpha_t\right)\sqrt{1 - \bar{\alpha}_t}}{\left(1 - \bar{\alpha}_t\right)\sqrt{\alpha_t}}\epsilon_0 \\
&= \left(\frac{\alpha_t\left(1 - \bar{\alpha}_{t-1}\right)}{\left(1 - \bar{\alpha}_t\right)\sqrt{\alpha_t}} + \frac{1 - \alpha_t}{\left(1 - \bar{\alpha}_t\right)\sqrt{\alpha_t}}\right)y_t - \frac{1 - \alpha_t}{\sqrt{1 - \bar{\alpha}_t}\sqrt{\alpha_t}}\epsilon_0 \\
&= \frac{\alpha_t - \bar{\alpha}_t + 1 - \alpha_t}{\left(1 - \bar{\alpha}_t\right)\sqrt{\alpha_t}}y_t - \frac{1 - \alpha_t}{\sqrt{1 - \bar{\alpha}_t}\sqrt{\alpha_t}}\epsilon_0 \\
&= \frac{1 - \bar{\alpha}_t}{\left(1 - \bar{\alpha}_t\right)\sqrt{\alpha_t}}y_t - \frac{1 - \alpha_t}{\sqrt{1 - \bar{\alpha}_t}\sqrt{\alpha_t}}\epsilon_0 \\
&= \frac{1}{\sqrt{\alpha_t}}y_t - \frac{1 - \alpha_t}{\sqrt{1 - \bar{\alpha}_t}\sqrt{\alpha_t}}\epsilon_0.
\end{aligned}
\tag{18}
$$

Therefore, we can set our approximate denoising transition mean $\mu_\theta(y_t, t \mid x, a)$ as

$$
\mu_\theta\left(y_t, t \mid x, a\right) = \frac{1}{\sqrt{\alpha_t}}y_t - \frac{1 - \alpha_t}{\sqrt{1 - \bar{\alpha}_t}\sqrt{\alpha_t}}f_\theta\left(y_t, t \mid x, a\right),
\tag{19}
$$

and the corresponding optimization problem becomes

$$\arg\min_{\theta} D_{\mathrm{KL}}\left(q\left(y_{t-1} \mid y_t, y_0\right) \| p_\theta\left(y_{t-1} \mid y_t, x, a\right)\right)$$

$$=\arg\min_{\theta} D_{\mathrm{KL}}\left(\mathcal{N}\left(y_{t-1}; \mu_q, \Sigma_q(t)\right) \| \mathcal{N}\left(y_{t-1}; \mu_\theta, \Sigma_q(t)\right)\right)$$

$$=\arg\min_{\theta} \frac{1}{2\sigma_q^2(t)}\left[\left\|\frac{1}{\sqrt{\alpha_t}}y_t - \frac{1-\alpha_t}{\sqrt{1-\bar{\alpha}_t}\sqrt{\alpha_t}}f_\theta\left(y_t, t \mid x, a\right) - \frac{1}{\sqrt{\alpha_t}}y_t + \frac{1-\alpha_t}{\sqrt{1-\bar{\alpha}_t}\sqrt{\alpha_t}}\epsilon\right\|_2^2\right]$$

$$=\arg\min_{\theta} \frac{1}{2\sigma_q^2(t)}\left[\left\|\frac{1-\alpha_t}{\sqrt{1-\bar{\alpha}_t}\sqrt{\alpha_t}}\epsilon - \frac{1-\alpha_t}{\sqrt{1-\bar{\alpha}_t}\sqrt{\alpha_t}}f_\theta\left(y_t, t \mid x, a\right)\right\|_2^2\right]$$

$$=\arg\min_{\theta} \frac{1}{2\sigma_q^2(t)}\left[\left\|\frac{1-\alpha_t}{\sqrt{1-\bar{\alpha}_t}\sqrt{\alpha_t}}\left(\epsilon - f_\theta\left(y_t, t \mid x, a\right)\right)\right\|_2^2\right]$$

$$=\arg\min_{\theta} \frac{1}{2\sigma_q^2(t)}\frac{\left(1-\alpha_t\right)^2}{\left(1-\bar{\alpha}_t\right)\alpha_t}\left[\left\|\epsilon - f_\theta\left(y_t, t \mid x, a\right)\right\|_2^2\right].$$

(20)

## D  Proofs

**Theorem 2** (Neyman-orthogonality)**.** *The orthogonal diffusion loss*

$$\mathcal{L}(\theta, \hat{\pi}) = \mathbb{E}_{(y_0, x, a) \sim p(Y, X, A); \epsilon \sim \mathcal{N}(\mathbf{0}, \mathbf{I})} \left[ w_{\hat{\pi}}(x, a) \left\| \epsilon - f_\theta \left( \sqrt{\bar{\alpha}_t} y_0 + \sqrt{1 - \bar{\alpha}_t} \epsilon, t \mid x, a \right) \right\|^2 \right], \quad (21)$$

*is Neyman-orthogonal wrt. its nuisance functions.*

*Proof.* As it was shown in the Sec. 5.1, the minimization of the orthogonal diffusion loss is equivalent to the maximization of the following weighted ELBO wrt. to $\theta$:

$$\mathcal{E}(p_\theta, \hat{\pi}) = \mathbb{E}_{(y_0, x, a) \sim p(Y, X, A)} \left[ w_{\hat{\pi}}(x, a) \, \mathbb{E}_{y_{1:T} \sim q(Y_{1:T} | y_0)} \left[ \log \frac{p_\theta (y_{0:T} \mid x, a)}{q (y_{1:T} \mid y_0)} \right] \right]. \quad (22)$$

We denote a maximizer of the weighted ELBO with the ground truth nuisance functions by $p_{\theta^*} = \operatorname{argmax}_{p_\theta} \mathcal{E}(p_\theta, \pi)$. Given a flexible enough diffusion model, $p_{\theta^*}(y_0 \mid x, a)$ coincides with the ground truth posterior and, thus matches the ground truth conditional distribution, namely, $p(Y \mid X, A)$ [47].

To demonstrate Neyman-orthogonality, we need to show that a pathwise cross-derivative is equal to zero [18, 60, 44], i.e.,

$$D_\pi D_{p_{\theta^*}} \mathcal{E}(p_{\theta^*}, \pi)[p_\theta - p_{\theta^*}, \hat{\pi} - \pi] = 0 \quad \text{for every } p_\theta \text{ and } \hat{\pi}. \quad (23)$$

We start by taking the pathwise derivative wrt. the optimal diffusion model $p_{\theta^*}$:

$$D_{p_{\theta^*}} \mathcal{E}(p_{\theta^*}, \pi)[p_\theta - p_{\theta^*}] \quad (24)$$

$$= \frac{\mathrm{d}}{\mathrm{d}t} \mathbb{E}_{(y_0, x, a) \sim p(Y, X, A)} \left[ w_\pi(x, a) \, \mathbb{E}_{y_{1:T} \sim q(Y_{1:T} | y_0)} \left[ \log \frac{p_{\theta^*} (y_{0:T} \mid x, a) + t \big( p_\theta (y_{0:T} \mid x, a) - p_{\theta^*} (y_{0:T} \mid x, a) \big)}{q (y_{1:T} \mid y_0)} \right] \right] \Bigg|_{t=0} \quad (25)$$

$$= \mathbb{E}_{(y_0, x, a) \sim p(Y, X, A)} \left[ w_\pi(x, a) \, \mathbb{E}_{y_{1:T} \sim q(Y_{1:T} | y_0)} \left[ \frac{p_\theta (y_{0:T} \mid x, a) - p_{\theta^*} (y_{0:T} \mid x, a)}{p_{\theta^*} (y_{0:T} \mid x, a) + t \big( p_\theta (y_{0:T} \mid x, a) - p_{\theta^*} (y_{0:T} \mid x, a) \big)} \right] \right] \Bigg|_{t=0} \quad (26)$$

$$= \mathbb{E}_{(y_0, x, a) \sim p(Y, X, A)} \left[ w_\pi(x, a) \, \mathbb{E}_{y_{1:T} \sim q(Y_{1:T} | y_0)} \left[ \frac{p_\theta (y_{0:T} \mid x, a) - p_{\theta^*} (y_{0:T} \mid x, a)}{p_{\theta^*} (y_{0:T} \mid x, a)} \right] \right]. \quad (27)$$

Then, we take a derivative wrt. the propensity score $\pi$:

$$D_\pi D_{p_{\theta^*}} \mathcal{E}(p_{\theta^*}, \pi)[p_\theta - p_{\theta^*}, \hat{\pi} - \pi] \tag{28}$$

$$=\frac{\mathrm{d}}{\mathrm{d}t} \mathbb{E}_{(y_0,x,a)\sim p(Y,X,A)} \left[ w_{\pi+t(\hat{\pi}-\pi)}(x,a) \, \mathbb{E}_{y_{1:T}\sim q(Y_{1:T}|y_0)} \left[ \frac{p_\theta(y_{0:T} \mid x, a) - p_{\theta^*}(y_{0:T} \mid x, a)}{p_{\theta^*}(y_{0:T} \mid x, a)} \right] \right] \Bigg|_{t=0} \tag{29}$$

$$=\mathbb{E}_{(y_0,x,a)\sim p(Y,X,A)} \left[ \left( -\frac{a}{(\pi(x))^2} + \frac{1-a}{(1-\pi(x))^2} \right) \mathbb{E}_{y_{1:T}\sim q(Y_{1:T}|y_0)} \left[ \frac{p_\theta(y_{0:T} \mid x, a) - p_{\theta^*}(y_{0:T} \mid x, a)}{p_{\theta^*}(y_{0:T} \mid x, a)} \right] \right] \tag{30}$$

$$=\mathbb{E}_{x\sim p(X)} \left[ -\frac{p(A=1 \mid x)}{(\pi(x))^2} \, \mathbb{E}_{y_0\sim p(Y|x,1)} \mathbb{E}_{y_{1:T}\sim q(Y_{1:T}|y_0)} \left[ \frac{p_\theta(y_{0:T} \mid x, 1) - p_{\theta^*}(y_{0:T} \mid x, 1)}{p_{\theta^*}(y_{0:T} \mid x, 1)} \right] \right. \tag{31}$$

$$\left. + \frac{p(A=0 \mid x)}{(1-\pi(x))^2} \, \mathbb{E}_{y_0\sim p(Y|x,0)} \mathbb{E}_{y_{1:T}\sim q(Y_{1:T}|y_0)} \left[ \frac{p_\theta(y_{0:T} \mid x, 0) - p_{\theta^*}(y_{0:T} \mid x, 0)}{p_{\theta^*}(y_{0:T} \mid x, 0)} \right] \right] \tag{32}$$

$$=\mathbb{E}_{x\sim p(X)} \left[ -\frac{1}{\pi(x)} \int \frac{p_\theta(y_{0:T} \mid x, 1) - p_{\theta^*}(y_{0:T} \mid x, 1)}{p_{\theta^*}(y_{0:T} \mid x, 1)} \, p(y_0 \mid x, 1) \, q(y_{1:T} \mid y_0) \, \mathrm{d}y_0 \, \mathrm{d}y_{1:T} \right. \tag{33}$$

$$\left. + \frac{1}{1-\pi(x)} \int \frac{p_\theta(y_{0:T} \mid x, 0) - p_{\theta^*}(y_{0:T} \mid x, 0)}{p_{\theta^*}(y_{0:T} \mid x, 0)} \, p(y_0 \mid x, 0) \, q(y_{1:T} \mid y_0) \, \mathrm{d}y_0 \, \mathrm{d}y_{1:T} \right] \tag{34}$$

$$\overset{(*)}{=}\mathbb{E}_{x\sim p(X)} \left[ -\frac{1}{\pi(x)} \int \frac{p_\theta(y_{0:T} \mid x, 1) - p_{\theta^*}(y_{0:T} \mid x, 1)}{p_{\theta^*}(y_{1:T} \mid y_0, x, 1)} \, q(y_{1:T} \mid y_0) \, \mathrm{d}y_0 \, \mathrm{d}y_{1:T} \right. \tag{35}$$

$$\left. + \frac{1}{1-\pi(x)} \int \frac{p_\theta(y_{0:T} \mid x, 0) - p_{\theta^*}(y_{0:T} \mid x, 0)}{p_{\theta^*}(y_{1:T} \mid y_0, x, 0)} \, q(y_{1:T} \mid y_0) \, \mathrm{d}y_0 \, \mathrm{d}y_{1:T} \right] \tag{36}$$

$$\overset{(**)}{=}\mathbb{E}_{x\sim p(X)} \left[ -\frac{1}{\pi(x)} \int \left( p_\theta(y_{0:T} \mid x, 1) - p_{\theta^*}(y_{0:T} \mid x, 1) \right) \mathrm{d}y_0 \, \mathrm{d}y_{1:T} \right. \tag{37}$$

$$\left. + \frac{1}{1-\pi(x)} \int \left( p_\theta(y_{0:T} \mid x, 0) - p_{\theta^*}(y_{0:T} \mid x, 0) \right) \mathrm{d}y_0 \, \mathrm{d}y_{1:T} \right] \tag{38}$$

$$=\mathbb{E}_{x\sim p(X)} \left[ -\frac{1}{\pi(x)}(1-1) + \frac{1}{1-\pi(x)}(1-1) \right] = 0, \tag{39}$$

where the equality $(*)$ holds due $p(y_0 \mid x, a) = p_{\theta^*}(y_0 \mid x, a)$; and the equality $(**)$ holds due to that the forward process is independent of $x$ and $a$, namely, $p_{\theta^*}(y_{1:T} \mid y_0, x, a) = p_{\theta^*}(y_{1:T} \mid y_0) = q(y_{1:T} \mid y_0)$. $\qquad \square$

# E Implementation details

In the following, we summarize the implementation details of our DiffPO and baselines.

## E.1 Implementation details of DiffPO

We implemented our DiffPO in PyTorch. (Code is available at `https://github.com/yccm/DiffPO`). Experiments were carried out on 1× NVIDIA A100-PCIE-40GB. We report the default settings of our model below but note that hyperparameters may require slight adjustments depending on the dataset.

Our model architecture of the denoising function $f_\theta$ is based on the architecture of [59], and we use U-Net [49] as the basic backbone. We use 4 diffusion residual blocks, where each is built with MLP layers. The diffusion embedding dimension is 128. The $\beta$ starts at 0.0001 and ends at 0.5, and the schedule is the "quadratic" version. The number of diffusion sampling steps is set to 100. The training batch size is set to 256 with a learning rate of 0.0005. The training epoch is set to 500.

During training, we can only observe one of the two POs due to the fundamental problem of causal inference [27] (as explained in Sec. 4). To guide the model, we need to identify which of two POs should be generated. Hence, we introduce causal masks as input to our model: an observational mask $m_o$, a targeted mask $m_t$, and a conditional mask $m_c$. For the observational mask $m_o$, it is 1 at the place where we have the observational data and 0 for the opposite case. Since we condition on $x$ and $a$, the conditional mask $m_c$ is 1 for the element $x$ and $a$, and 0 otherwise. For the target mask, the observed outcomes $y$ in the original dataset are 1, while all the other elements are 0. In this way, we can compute the loss only at the place where the value of the targeted mask is 1, i.e., where we have the ground truth outcomes. We follow [42] in the way how we learn propensity scores and use fully connected neural networks with softmax activation. Thus we add weight to each sample via a learned function when computing the orthogonal loss.

## E.2 Implementation details of baselines

We follow the implementation from `https://github.com/AliciaCurth/CATENets/tree/main` for most of the CATE estimators, including S-learner [37], T-learner [37], DR-learner [39, 31], RA-learner [12], TARNet [54]. For GANITE, we follow the implementation of `https://github.com/vanderschaarlab/mlforhealthlabpub/tree/main/alg/ganite`. For TEDVAE, we follow the implementation of `https://github.com/WeijiaZhang24/TEDVAE`.

We performed hyperparameters tuning of the nuisance functions models for all the baselines based on five-fold cross-validation using the training subset. For each baseline, we performed a grid search with respect to different tuning criteria, evaluated on the validation subsets. We aimed for a fair comparison and thus kept the number of parameters, network structures, and grid size similar across models. For evaluating the uncertainty estimation, for both training and testing, the dropout probabilities were set to $p = 0.1$.

# F  Dataset details

## F.1  Synthetic dataset

We use one of the datasets from ACIC2018 with 177 covariates to generate the synthetic dataset with a coefficient matrix $W$, i.e.,

$$\begin{cases} X \sim \text{ Real-World }(\cdot), \\ A \sim \text{ Real-World }(X), \\ Y := U_Y + \sin(WX) + AX + A; \quad U_Y \sim N(0,1) \end{cases} \tag{40}$$

We use a ten-fold split for train/test samples $(80\%/20\%)$.

## F.2  ACIC 2016 & 2018 datasets

The 2016 Atlantic Causal Inference Challenge (ACIC2016) [46] contains 77 different settings of benchmark datasets, and ACIC2018 [41] contains 24, respectively. They are designed to benchmark causal inference algorithms with various data-generating mechanisms. Covariates of ACIC 2016 are taken from a large study of developmental disorders, and covariates of ACIC 2018 are derived from the linked birth and infant death data. ACIC 2016 and ACIC 2018 differ in the number of true confounders, the varying levels of overlap, and the form of conditional outcome distributions. ACIC 2016 has 77 different data-generating mechanisms with 100 equal-sized samples for each mechanism $(n = 4802, d_X = 82)$. ACIC 2018 provides 63 distinct data-generating mechanisms with around 40 non-equal-sized samples for each mechanism ($n$ ranges from $1,000$ to $50,000$, $d_X = 177$). Notably, ACIC 2018 has a constant CATE for most of the datasets, but heterogeneous propensity scores.

## F.3  IHDP dataset

The Infant Health and Development Program (IHDP) [25] has features and treatment assignments from a real-world clinical trial. The dataset comprised 747 subjects, with 25 features for each subject. Out of the 25 features, 6 are continuous, and 19 are binary with a binary treatment. For both treated and untreated, synthetic outcomes of IHDP are sampled from different conditional normal distributions. These distributions are homoscedastic ($\sigma^2 = 1$) but have substantially different conditional means. The potential outcomes are simulated according to the standard non-linear "Response Surface B" setting in [25] with the following data-generating mechanism:

$$\begin{cases} X \sim \text{Real-World}(\cdot), \\ A \sim \text{Real-World}(X), \\ Y \sim N\big(A\,(X\beta - \omega) + (1 - A)\,(\exp((X + W)\beta)), 1\big), \end{cases} \tag{41}$$

where $\beta, W, \omega$ are constant parameters of the simulation. For further details, we refer to [25].

# G   Additional experiments

## G.1   Simulation experiments

We examine the Neyman-orthogonality property of our orthogonal diffusion loss through simulation experiments. We conduct experiments to show that the orthogonal diffusion loss can address the problem of model misspecifications when estimating nuisance functions (e.g., the propensity score $\pi(x)$).

To this end, we perturb the propensity score manually to assess the robustness of the loss when the nuisance functions are estimated with varying errors. During training, we consistently increase the sample size and evaluate the CATE estimation error on the synthetic dataset. As shown in Fig. 3, the CATE estimation error decreases as the sample size grows and eventually converges to zero. This demonstrates that, even when the propensity score is misspecified, the orthogonal loss remains robust, enabling our DiffPO to converge to the correct objective. This experiment supports Theorem 1 by illustrating how the stability and consistency of the orthogonal loss contribute to convergence. In sum, it shows the theoretical benefits of our proposed orthogonal diffusion loss.

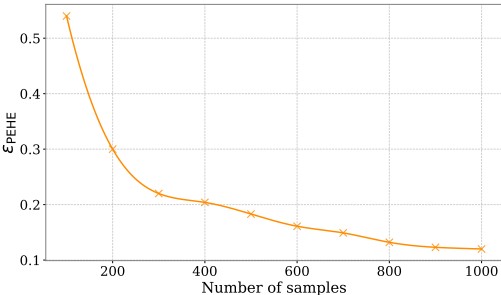

Figure 3: We manually perturb the propensity score during training on the synthetic data. We replace the estimated propensity score $\hat{\pi}(x)$ with a randomly sampled value $\tilde{\pi}(x)$ from the interval $(0, 1)$. The weight $w_{\hat{\pi}}(x, a)$ for each sample in the orthogonal diffusion loss $\mathcal{L}(\theta, \hat{\pi})$ is thus replaced by weight $w_{\tilde{\pi}}(x, a)$. The CATE estimation error gradually converges as the sample size increases. This aligns with our expectation, as the loss remains robust even with varying errors in the estimation of nuisance functions.

