# OpenReview forum: "DiffPO: A causal diffusion model for learning distributions of potential outcomes"
_NeurIPS.cc/2024/Conference — NeurIPS 2024 poster_

### Official Review · Reviewer_Cvxf · 2024-07-11

**Soundness:** 3
**Presentation:** 3
**Contribution:** 3
**Rating:** 6
**Confidence:** 4

**Summary:**

This paper proposes DiffPO, a causal diffusion model, to predict individual potential outcomes. The authors motivate why predicting potential outcomes can be a more complex problem than (conditional) average treatment effect (CATE) prediction in practical settings. They adapt a conditional denoising diffusion model for this problem setting and augment its training with an orthogonal diffusion loss to correct for confounding. The proposed method is applicable to diverse settings and empirical evaluation demonstrates performance superior to comparable baselines across datasets. Further, the method is deployed to measure CATE, which is direct function of the predicted potential outcomes. The diffusion-model based instantiation of this method lends itself readily to uncertainty quantification, which indicates greater reliability of the method in high-stakes domains like medicine.

**Strengths:**

Originality: The motivation of the paper is particularly interesting. It is intuitive that a method designed for good ATE/CATE prediction will not necessarily perform well in potential outcome prediction. The paper motivates the need to separately focus on the latter setting well.

Quality: Experimental evaluation of the method is quite diverse and extensive, demonstrating superior performance on various datasets compared to baselines.

Clarity: The motivation, method and overall paper are well-written and easy to follow.

Significance: The paper highlights and addresses significant aspects of causal inference in medical applications, like individual effects, and uncertainty quantification.

**Weaknesses:**

Clarity:

- I found the preliminaries about diffusion models highly compressed (which is understandable due to space constraints). Since the proposed method relies heavily on diffusion modeling, the authors might consider using the extra space allowed in camera ready versions for expanding on the relevant details about training diffusion models.

- Authors should also make sure to reference all tables and figures in the main text to better guide the reader to results relevant to a given section. Nit: line 309 does have what was probably intended as a reference but is broken.

**Questions:**

- The three different kind of experiments (PO estimation, CATE estimation, PO distribution estimation) all use different datasets. Is there a reason all these quantities cannot be estimated for all datasets?

- How sensitive is the proposed estimator to hyperparameter, choice of model architecture or other training details?

**Limitations:**

The paper discusses limitations due to assumptions and efficiency of the method.

---

> ### Author Rebuttal · Authors · 2024-08-07
>
> Thank you for your positive review and your helpful comments! We appreciate that you find our paper interesting, important, and comprehensive. We are very delighted to answer your questions and improve our paper as a result.
>
> ## Response to “Weaknesses”
>
> * Thank you very much for your suggestions! Yes, we compressed the preliminaries to save space. **Action:** We will add more details about our method in order to make it more easy to apply. We will also use the extra space in the camera-ready version to expand our section with implementation details to provide more information about how we train the diffusion models.
>
> * Thank you! **Action:** We will fix the typos and make sure that all tables and references are properly cross-referenced in the camera-ready version.
>
>
> ## Answer to “Questions”:
>
> **(Q1) Choice of datasets**
>
> Thank you for asking this question! Actually, there is no specific reason for splitting the estimation of these causal quantities on different datasets. Our rationale was simply that we wanted to empirically demonstrate the flexibility and generalization power of our model across a broad range of datasets.
>
> However, some commonly used datasets indeed can **not** be used for evaluating all three causal quantities. For example,  ACIC2016 (with 77 datasets) and ACIC2018 (with 24 datasets) datasets, which we also used in our paper. These datasets are designed for CATE benchmarking. It is impossible to use it for some other tasks, such as evaluating distribution. This is because these datasets only provide a single sample from the distribution instead of the full distribution knowledge.  Thus the ground truth is unavailable.
>
> Therefore, for CATE estimation, we benchmarked on ACIC2016 and ACIC2018. We additionally report the average performance on ACIC2016 and ACIC2018   (see **Table 2/Rebuttal PDF**). For evaluating distribution, we report the experiment results using the empirical Wasserstein distance on the semi-synthetic dataset where the ground truth distribution is available (see Table 1 / Rebuttal PDF). We find that we outperform the baselines by a clear margin for both tasks. The results also demonstrate our method is highly effective and robust across various datasets.
>
> **Action:** Previously, we have presented only a selection of our experiments in the main paper due to space constraints. In the camera-ready version, we will report additional experiments to streamline the presentation.
>
> **(Q2) Robustness to hyperparameters**
>
> Thank you for asking this question! We found that our proposed model is robust and not very sensitive to the hyperparameters and model architecture. We performed a grid search of hyperparameters in a reasonable range. For example, even when changing various parameters (e.g., number of neurons), the overall performance was state-of-the-art and stable. One factor that, however, was relevant was the use of the clipping propensity score technique to avoid the numerical instability of the propensity score during training, yet this technique is commonly also applied to the baselines in order to improve their robustness.
>
> **Action:** We will report our tuning grid during hyperparameter tuning in our revised paper, and we will further discuss that our method is largely robust to different choices of the hyperparameters.

---

> > ### Comment · Reviewer_Cvxf · 2024-08-10
> >
> > Thank you for the clarifications and additional results! I will keep my score.

---

### Official Review · Reviewer_PmLo · 2024-07-11

**Soundness:** 2
**Presentation:** 2
**Contribution:** 2
**Rating:** 5
**Confidence:** 4

**Summary:**

This paper aims to develop a model which focuses on potential outcomes (POs) instead of CATE. It focuses on estimation of POs and proposed an estimation method which is an extension of the well know diffusion model with a different loss function.

**Strengths:**

- This work proposes a diffusion model based estimation method for potential outcome estimation. The method is a simple extension of the standard diffusion model.

**Weaknesses:**

- The motivation behind this is not clear as in most of the cases people are interested in the causal effect instead of potential outcomes, especially given one potential outcome is observed already.
- The main novelty of this work is the orthogonal diffusion loss. However, it is simply using the idea of inverse propensity scores.
- There is nothing specially designed for potential outcomes in the forward and reverse process, the variational inference and the conditional denoising function, which should not be clarified as background instead of methodology.

----
details
- L350 In fact any variational inference based causal inference method can predict the distribution of potential outcomes. In addition, conformal prediction based causal inference methods can predict a predictive interval for each potential outcome.
- Eq(11) It is unclear what the expectation is taken over.
- L189 it is not a sentence.
- L118 I don't think those papers are totally unrelated from this work.
- Table 1. I believe for the purpose of this work, the comparison should be between different estimation models. The meta learners should be applicable along with the proposed model and other baselines, so it is not that crucial to list them in Table 1.

**Questions:**

- Remark 1 may need more discussion. How to prove the loss is Neyman-orthogonal? What is the benefit to make the loss Neyman-orthogonal? Is there any experiment showing the advantage of this loss?

**Limitations:**

See weakness

---

> ### Author Rebuttal · Authors · 2024-08-07
>
> Thank you for your review and your helpful comments!
>
>
> ## Weaknesses
>
> **(W1) Relevance of potential outcomes**
> - We would like to emphasize that potential outcomes are highly relevant in many decision-making settings such as medicine [1]. Example: predicting the survival probability under treatment A and the survival probability under treatment B, for a patient _with no previous treatment_. Here, predicting potential outcomes is directly relevant for clinical decision support** and informs which treatment is preferred (in terms of the health benefit but also side effects and quality of life) [1].
>
> - We would like to clarify that _one potential outcome is only observed in the training data, but **not** upon deployment_. Instead, **both potential outcomes must be predicted upon deployment** (aka in the training data).
>
> [1] Feuerriegel S, Frauen D, Melnychuk V, et al. Causal machine learning for predicting treatment outcomes. Nature Medicine, 2024.
>
>
> **(W2) IPS & Neyman-orthogonality**
>
> We admit that we leveraged the idea of inverse propensity score to address the selection bias. Nevertheless, we would like to clarify that our orthogonal diffusion loss for diffusion models has a clear advantage over existing literature in models for learning potential outcome distributions. In particular, **our orthogonal diffusion loss is Neyman-orthogonal wrt. the nuisance functions**. As such, we offer a theoretical guarantee that our loss is robust against errors in nuisance function estimation. This goes beyond the standard inverse propensity score from the literature, and the corresponding proof of Neyman-orthognality is _non-trivial_.
>
> To improve the rigor of our work, **we added theory and new experiments** for our novel orthogonal diffusion loss:
>
> 1. **Theoretical proof**:  **We provide the formal proof that our proposed orthogonal diffusion loss is Neyman-orthogonal wrt. its nuisance functions** (see **Theorem and Proof in the Rebuttal PDF**).
>
>
> 2. **New experiment**: **We include the ablation study of our orthogonal diffusion loss** where we compare against a vanilla diffusion model without our orthogonal diffusion loss. The experiment results allow us to assess the contribution of our novel orthogonal diffusion loss to the overall performance (see **Table 4 / Rebuttal PDF**). We observe a clear performance gain from including our novel loss.
>
> **Action**: We will add our theorem and our detailed proof of Neyman-orthogonality to our revised paper. We will also add the new experiments with our ablation to our revised paper in order to show the performance gain from our novel orthogonal diffusion loss.
>
>
> **(W3) Presentation of background and our novel loss**
>
> Thank you for your comment. It is true that methods based on conditional diffusion models always contain the forward and reverse process etc, regardless of whether the methods come from computer vision or other domains. Analogously, our method also adopts the forward and reverse process as in conditional diffusion models. However, our objective was **not** to change the forward and reverse process. Rather, **our objective was to develop a novel method for causal tasks**, and, for this reason, **we propose a novel orthogonal diffusion loss with theoretical properties** (see **Theorem and Proof in the Rebuttal PDF**).
>
> We realized that some parts of our method are familiar to readers with a background in diffusion models. However, readers from causal inference may also appreciate the background (see, e.g., Reviewer Cvxf). However, we understand that our presentation could be improved. **Action:** We will revise our paper and split the method section into two sections presenting ‘what is known’ (=the diffusion model background) and ‘our innovation’ (=our novel orthogonal diffusion loss).
>
>
> ### Details
>
>
> **(Q1)**
>
> _Difference / variational inference methods_
>
> In principle, yes, any variational inference-based causal inference method can learn the distribution; however, none of the previous work has done so. Importantly, many of them use the variational-based method simply to learn latent variables but _then predict a point estimation of the CATE_ (e.g., TEDVAE).  In contrast, **we are the first variational inference-based causal inference method designed to learn the distribution of potential outcomes**, and **our method offers favorable theoretical properties (Neyman-orthogonality)**.
>
> _Difference / conformal prediction_
>
> It is true that conformal prediction (CP)-based causal inference methods can predict a predictive interval. However, we would like to point out the main difference: **Our method can learn the whole distribution but CP can not**. We also would like to clarify the **two main advantages of our method** over CP:
>
> 1. **More flexible.** CP targets the total uncertainty and needs to be calibrated each time for different levels of uncertainty. However, our method targets the aleatoric uncertainty and yields the full distribution of the potential outcomes at once.
>
> 2. **More general/useful.** There are cases in which the mean and the variance are the same for the outcomes, but the distribution of the outcomes is different. This kind of information is very useful and can be obtained from our method. However, CP can not provide such information.
>
> **(Q2)** Thank you. The expectation in Eq (11) is taken over $(y_0, x, a) sampled from observational distribution and $\epsilon$ sampled from normal distribution. We will add this to Eq (11) in our paper.
>
> **(Q3)** Thanks. We will fix the sentence.
>
> **(Q4)** Thanks. We will remove the term “unrelated” and use a more nuanced wording to say that the papers study a _different task_.
>
> **(Q5)** Thanks. We will rearrange Table 1 as per your suggestion.
>
>
> ## Questions
>
> Thank you for the suggestion. **We now provide the formal proof that our proposed orthogonal diffusion loss is Neyman-orthogonal wrt. its nuisance functions** (see **Theorem and Proof in the Rebuttal PDF**).

---

> > ### Comment · Reviewer_PmLo · 2024-08-13
> >
> > Thanks for the very helpful response. I raised my score to 5.

---

> > > ### Author Response · Authors · 2024-08-13
> > > **Response to Reviewer PmLo**
> > >
> > > Thank you so much for your response and for raising the score!
> > >
> > > We will include all suggested points in the paper.
> > > Thank you again for your effort in reviewing this paper and your constructive comments!

---

### Official Review · Reviewer_KUb5 · 2024-07-13

**Soundness:** 3
**Presentation:** 3
**Contribution:** 3
**Rating:** 4
**Confidence:** 4

**Summary:**

This paper applies the technology of diffusion models to estimation of potential outcome and treatment effect. Owing to the capability of modelling distribution in diffusion model, this paper can not only give point estimation but also uncertainty. Specifically, it proposes to use variational inference to learn the model and applies sample re-weighting to remove confounding bias. The estimation is proved to satisfiying Neyman-orthogonality property, and empirical validation reveal the effectiveness of the method.

**Strengths:**

This paper exploits the power of diffusion models and achieve more capability than previous works on CATE estimation. The proposed method is technologically sound and achieve Neyman-orthogonality, which is an important theoretical property. The experimental results demonstrate the effectiveness of the method.

**Weaknesses:**

Although the paper is  technologically sound and well-written, the empirical examination is somewhat lacking. There have been a lot of dataset, such IHDP, Jobs, Twins in CATE estimation. I suggest the authors to complement the experimental results on these datasets to enhance the persuasive appeal of the experiments. There are additional questions about the experiments, see the details in Questions.

**Questions:**

1. The results in Table 3 (CATE estimation) is confusing. The authors claim the metric is PEHE, which is favored to be lower. However, the proposed method (DiffPO) achieves largest value in Table 3. What does the value in Table 3 mean?
2. The authors examines the inferred potential outcome distribution in Section 6.3. However, how to validate the rationality of the inferred distribution? I think it is a very important problem. Otherwise, the inferred results are not trustworthy.

**Limitations:**

The authors have clearly listed the limitation of the paper, and claim the potential solution to address these limitations.

---

> ### Author Rebuttal · Authors · 2024-08-07
>
> Thank you for your review and your helpful comments! We appreciate that you find our proposed method important, sound, and effective. We further would like to emphasize that we perform CATE experiments to show that our method is flexible. Nevertheless, the primary objective of our method is to _learn the distribution of potential outcomes_. We are happy to answer your questions and improve our paper as a result.
>
> ## Response to Weaknesses:
>
> We thank you for your suggestion to add more datasets. **We additionally conducted experiments for the three datasets that you suggested: IHDP, Jobs, and Twins** (see **Table 3 / Rebuttal PDF**). We find that our method is comparable to existing CATE methods and achieves state-of-the-art performance.
>
> **Action**: We will add the above experiments to our revised paper.
>
> ## Answer to “Questions”:
>
> **(Q1) Question about whether lower/higher PEHE is preferred**
>
> We thank the reviewer for the important question. Yes, you are right that a **lower** value is favored for the PEHE evaluation metric (Table 1 in our original submission). In contrast, Table 3 in our original submission does not report PEHE but  “% of runs with the best performance”, but where a **higher** value is preferred. The reason for the choice of the evaluation metric is that ACIC2016 and ACIC2018 are ‘suites’ with multiple benchmark datasets; that is, they contain 77 different and 24 different benchmark datasets, respectively.
>
> Upon reading your question, we realized that we should evaluate ACIC2016 and ACIC2018 not only in terms of “% of runs with the best performance” (where higher is better) but also in terms of average PEHE (where lower is better). Hence, **we repeated our experiments and now report the average PEHE** for these experiments (see **Table 2 / Rebuttal PDF**). Here, lower values in terms of average PEHE are favored (consistent with Table 1 from our original submission). Overall, **our proposed method is better by a clear margin**. For example, on ACIC2016, the best baseline has an out-of-sample average PEHE of 0.65, while our method achieves 0.49. On ACIC2018, the best baseline has an out-of-sample average PEHE of 0.28, while our method achieves 0.14. This corresponds to improvements of 24.6% and 50.0%, respectively.
>
> We also manually inspected where our method was superior to offer qualitative insights. Generally, we found that our method was very strong and overall best (see our "% of runs with best performance" in the original paper). Nevertheless, we observed a tendency that, for datasets with really simple data-generating mechanisms (for example, all linear), our method may not be best, and other, simpler models are better.
>
> **Action:** We will add the above results to our revised paper.
>
> **(Q2) New experiments to evaluate the learned distribution**
>
> Thank you for asking this important question. **We performed new experiments where we explicitly assessed the ability of our method to learn the distribution of potential outcomes.**  We thus compare the distributions using the empirical Wasserstein distance. We find that we outperform the baselines by a clear margin (see **Table 1 / Rebuttal PDF**).
>
>
> **Action:** We will add the above experiments to our revised paper.

---

> > ### Comment · Reviewer_KUb5 · 2024-08-13
> >
> > Thanks for your response and effort in revise paper. The response address almost all my concerns. Still, I have another question. What does the empirical Wasserstein distance means in evaluating the learned distribution?

---

> > > ### Author Response · Authors · 2024-08-13
> > > **Response to Reviewer KUb5**
> > >
> > > Thank you for your reply! We are happy that we have addressed your concerns sufficiently!
> > >
> > > We also welcome the opportunity to answer your remaining question about the empirical Wasserstein distance.
> > >
> > > The Wasserstein distance is a commonly used metric in machine learning [1,2,3] for quantifying the distance between two probability distributions based on the concept of optimal transport. In our work, we use it to evaluate the model performance in learning the potential outcome distributions. Thus, this allows us to move _beyond evaluating point estimates_ to the _full distribution_.
> > >
> > >
> > > The $k$-Wasserstein distance (for any $k \geq 1$) between two distributions $\mu_1$ and $\mu_2$ is
> > >
> > > $W^k\left(\mu_1, \mu_2\right)=\left(\int_0^1\left|\mathbb{F}_1^{-1}(l)-\mathbb{F}_2^{-1}(l)\right|^k \mathrm{~d} l\right)^{1 / k}$
> > >
> > > where $\mathbb{F}_1^{-1}(l)$ and $\mathbb{F}_2^{-1}(l)$ are the quantile functions (inverse cumulative distribution functions) of $\mu_1$ and $\mu_2$ for quantile $l$, respectively.
> > >
> > > In practice, Wasserstein distance is estimated by the empirical Wasserstein distance based on two sets of finite samples, i.e.,
> > >
> > > $\hat{W}^k\left(\mu_1, \mu_2\right)$ $=\left(\frac{1}{n} \sum_{i=1}^n\left\|X_i-Y_i\right\|^k\right)^{1 / k}$,
> > >
> > > where $X_1, \ldots, X_n$ are the samples from $\mu_1$ and $Y_1, \ldots, Y_n$ are the samples from $\mu_2$, and we set $k=1$ in our experiments, following [1].
> > >
> > > (3) We use empirical Wasserstein distance to evaluate our model performance on learning the potential outcome distributions, where _lower_ distances imply _better_ performance. Here, our method gives the **best** results and outperforms the baselines by a clear margin (see **Table 1 / Rebuttal PDF**). This demonstrates the strong performance of our model not only on point estimates, but also on estimating the _full distribution_.
> > >
> > >
> > > Thank you again for reading our rebuttal! We hope that the above results answer all your questions.
> > >
> > > We would greatly appreciate it if you would raise your score accordingly. Thank you. And do let us know if you have further questions -- we will do our best to answer them promptly.
> > >
> > > Thanks again for reviewing our submission!
> > >
> > >
> > > Reference:
> > >
> > > [1]  Maeda T, Ukita N. Fast inference and update of probabilistic density estimation on trajectory prediction. International Conference on Computer Vision. 2023
> > >
> > > [2] Melnychuk V, Frauen D, Feuerriegel S. Normalizing flows for interventional density estimation. International Conference on Machine Learning. 2023.
> > >
> > > [3] Letzelter V, Perera D, Rommel C, et al. Winner-takes-all learners are geometry-aware conditional density estimators. International Conference on Machine Learning. 2024.

---

### Official Review · Reviewer_vD2m · 2024-07-20

**Soundness:** 3
**Presentation:** 4
**Contribution:** 4
**Rating:** 6
**Confidence:** 3

**Summary:**

This paper describes DiffPO, a causal diffusion model for predicting distributions of potential outcomes, as well as related causal quantities such as CATE estimates. DiffPO accounts for the confounding between covariates and outcomes through a simple weighting procedure, under standard potential outcomes assumptions.  The paper evaluates its performance in comparison to a variety of PO and CATE estimators over several standard benchmarks, including the ACIC 2016 and 2018 datasets and TCGA.

**Strengths:**

- The application of a diffusion approach to potential outcome and CATE estimation is useful, especially given the flexibility to handle a variety of problem settings.

- The paper's theoretical formulations of causal diffusion is novel, including the orthogonal diffusion loss and the implications for the DiffPO ELBO and simplified training objective.

- The experiments show strong performance compared to baseline approaches for CATE estimation on several standard causal inference benchmark datasets.

**Weaknesses:**

- The high level evaluation results (table 2 and table 3) are impressive.  However, it would be good include additional details on benchmark performance, perhaps in the appendix.  For example, these results tables show the "% of runs with best performance".  It would be insightful to see the underlying performance metrics as well, to understand whether the approach was "just a little better" or a lot better.  This would also help evaluate whether there are contexts where DiffPO is consistently stronger or weaker than alternatives.

- The weighting approach in equation 10 will be unstable when samples have propensities approaching 0 or 1. While this is covered by standard causal assumptions related to population overlap, in practice it is common to have to clip propensity scores or drop high or low propensity samples. Was that unnecessary here, or what approach was chosen?

**Questions:**

Does this approach lend itself to sensitivity analyses (e.g., of the assumption that there are no latent confounders?)

Could you please define best_in and best_out in Tables 3 and 4?

I believe there are usually annual ACIC challenges.  Why were acic 2016 and acic 2018 chosen over other years benchmarks?

The paper claims DiffPO can adapt to continuous treatments --- but the experiments seem to all be binary treatments? Did I miss something about the ACIC datasets or were continuous treatments not evaluated?

**Limitations:**

Limitations were adequately discussed though, I think the claim (in the limitations paragraph, line 366) that the method scales to real-world datasets based on ACIC, IHDP, and TCGA is perhaps a little much to claim given the breadth of real-world scenarios and datasets, and the fact that these are datasets created or collated for academic purposes.

---

> ### Author Rebuttal · Authors · 2024-08-07
>
> Thank you for your positive review and your helpful comments! We appreciate that you find our paper novel, useful, and with strong experiment performance. We are very happy to answer your questions and improve our paper as a result.
>
>
> ## Response to “Weaknesses”
>
> **(W1) Additional performance merics for ACIC2016 & ACIC2018**
>
> Thank you for your suggestions! We now report the **average performance for PEHE metrics** on ACIC2016 (77 datasets) and ACIC 2018 (24 datasets) (see **Table 2 / Rebuttal PDF**). Here, we compute the average of PEHE across all the datasets (lower values are favored). Overall, **our proposed method is better by a clear margin**. For example, on ACIC2016, the best baseline has an out-of-sample average PEHE of 0.65, while our method achieves 0.49. On ACIC2018, the best baseline has an out-of-sample average PEHE of 0.28, while our method achieves 0.14. This corresponds to improvements of 24.6% and 50.0%, respectively.
>
> We also manually inspected where our method was superior to offer qualitative insights. Generally, we found that our method was very strong and overall best (see our "% of runs with best performance" in the original paper). Nevertheless, we observed a tendency that, for datasets with really simple data-generating mechanisms (for example, all linear), our method may not be best, and other, simpler models are better.
>
> **Action**: We will add the above results to our revised paper. We will further report a breakdown by dataset to allow for further qualitative insights.
>
> **(W2) Weighting / clipping**
>
> Thank you for asking this important question. Yes, you are absolutely right. For the datasets with poor overlap, extreme propensity scores (near 0 or 1) can lead to high variance. As written in your comment, a common approach is the clipping technique based on the propensity score to avoid numerical instability during training. We also use clipping in our implementation and set a minimum ($\epsilon$) and maximum  ($1- \epsilon$) threshold with $\epsilon = 0.05$.
>
> **Action:** We will add details about how we follow best-practice and use clipping to the implementation details of our revised paper.
>
> ## Answer to “Questions”
>
> 1. Thank you for asking this interesting question. That could be a very interesting extension of our current work, as, generally, sensitivity models require fitting the conditional distribution models as nuisance functions [5]. Here, a diffusion model like ours can serve as an estimator of the conditional distribution. While this is an interesting extension for future research, our current paper focuses on the setting, where unconfoundedness holds. **Action:** We will state in our revised paper that extensions of causal diffusion models to sensitivity analysis are an interesting opportunity for future research.
>
>
> 2. The "In-sample" and "out-of-sample" refer to the portion of the dataset that is used to train and evaluate the model, respectively. The “best_in” and “best_out” mean the “% of runs with the best performance” on the training and test dataset, respectively. **Action:** We will clarify the definition in the experiment section of our revised paper.
>
>
> 3. Yes, you are right that there are usually annual ACIC challenges. However, _these were mostly designed for other tasks/purposes rather than CATE estimation. Only ACIC2016 and ACIC 2018 are designed for benchmarking CATE, which is the reason why the two datasets are commonly used by previous works for CATE benchmarking [1,2,3,4].
>
>
> 4. Thank you. Our method can be adapted to continuous treatment settings but, upon closer inspection, the performance gains are not as clear as for the binary treatment. **Action:** We decided to remove that claim from our paper and, instead, focus on the true strength of our method: DiffPO is carefully designed for learning the _conditional distribution_ of potential outcomes.
>
> ## Response to “Limitations”:
> Thank you for your honest feedback! **Action:**  We will revise our statements in the camera-ready version of our paper to be more nuanced and toned down.
>
>
>
> Reference:
>
> [1] Mahajan, D., Mitliagkas, I., Neal, B. and Syrgkanis, V., 2022. Empirical Analysis of Model Selection for Heterogeneous Causal Effect Estimation. ICLR.
>
> [2] Zhang, J., Jennings, J., Zhang, C. and Ma, C., 2023. Towards causal foundation model: on duality between causal inference and attention. ICML.
>
> [3] Zhang W, Liu L, Li J, 2021. Treatment effect estimation with disentangled latent factors. AAAI.
>
> [4] Cousineau, M., Verter, V., Murphy, S. A., & Pineau, J. 2023. Estimating causal effects with optimization-based methods: A review and empirical comparison. European Journal of Operational Research.
>
> [5] Frauen, Dennis, et al, 2024. A neural framework for generalized causal sensitivity analysis. ICLR.

---

> > ### Comment · Reviewer_vD2m · 2024-08-12
> >
> > Thank you for answering my questions.

---

### Author Rebuttal · Authors · 2024-08-07

Thank you very much for the constructive and positive evaluation of our paper and your helpful comments! We addressed all of them in the comments below and uploaded **additional results as a PDF file**.

Our **main improvements** are the following:

* **We provide a theoretical guarantee:**
We provide **a formal proof that our proposed orthogonal diffusion loss is Neyman-orthogonal wrt. its nuisance functions** (see **Theorem and Proof in the Rebuttal PDF**). As such, we offer a theoretical guarantee that our loss is robust against errors in nuisance function estimation. This thus gives a proper justification of why our loss for predicting the distribution of potential outcomes is effective and thus beneficial in practice.

* **We provide further extensive experiments:**

(1) We evaluate **the learned distributions of the potential outcomes** (see **Table 1 / Rebuttal PDF**). We find that we outperform the baselines by a clear margin. Thereby, we show that our method is not only effective for providing point estimates but also for learning distributions.

(2) We include the **ablation study of our orthogonal diffusion loss** where we compare against a vanilla diffusion loss without our orthogonal diffusion loss. (see **Table 4 / Rebuttal PDF**). We observe a clear performance gain from including our novel loss (over a vanilla diffusion model). The results also support our theory (see **Theorem and Proof in the Rebuttal PDF**).

(3) We include **three additional CATE benchmarking datasets** and find our method gives state-of-the-art performance (see **Table 3 / Rebuttal PDF**).

(4) We report the **average performance** based on the PEHE evaluation metric **across ACIC2016 and ACIC 2018 datasets** (see **Table 2 / Rebuttal PDF**). The experiment results again demonstrate that our method is superior and robust.

As a summary, we would like to emphasize our contribution again: We develop a conditional denoising diffusion model to learn complex **distributions of potential outcomes**, where we address the selection bias through a **novel orthogonal diffusion loss**. We also offer a non-trivial, **theoretical proof that our orthogonal diffusion loss is Neyman-orthogonal** wrt. its nuisance functions. This gives theoretical support that our method is effective and why it outperforms the baselines.


We will incorporate all changes (labeled with **Action**) into the camera-ready version of our paper. Given these improvements, we are confident that our paper will be a valuable contribution to the machine learning for healthcare literature and a good fit for NeurIPS 2024.

---

### Decision · Program_Chairs · 2024-09-25

**Decision:**

Accept (poster)

**Comment:**

Overall, the reviewers appreciated the topic of the paper which formulates potential outcomes estimation as a diffusion problem and found the initial experiments convincing. Most of them appreciated the clarity of presentation. Multiple reviewers expressed concerns about the unclear connection to IPS and insufficient ablation studies and datasets. The authors addressed these concerns in the rebuttal and should update the manuscript accordingly.